# STABILITY-AWARE POST-TRAINING CASCADE OF EXPERTS FOR COMPUTE-EFFICIENT INFERENCE

## ABSTRACT

State-of-the-art models achieve high accuracy at the cost of substantial inference compute, hindering deployment on edge devices and under strict latency budgets. To address this, we present a stability-aware post-training cascade-of-experts that operates over a heterogeneous pool of pre-trained models, balancing accuracy, inference cost, and decision stability. Specifically, we address three questions: Which base models to select—from a heterogeneous pool we retain $\epsilon$-competitive candidates by $\epsilon$-Pareto screening the accuracy–compute trade-off, forming the cascade's candidate set; How to optimize thresholds—we learn stage-wise accept/defer thresholds via a recursive threshold-grid search with optimal tail-set reuse, minimizing expected inference cost subject to a user-set accuracy tolerance; What final cascade and execution order to deploy—we choose them by jointly considering expected inference cost and cross-validated decision stability. In experiments across text, vision, and audio, when the reference single model shows no substantial validation–test discrepancy, the framework delivers large compute reductions at a bounded accuracy drop.

## 1 INTRODUCTION

### 1.1 MOTIVATION

Modern models push accuracy upward but often inflate inference complexity, latency, and energy. This tension is acute on edge, mobile, and industrial hardware, and even in the cloud under strict service-level agreements (SLAs) on tail latency and cost. The community has developed a broad toolkit—compression (Han et al., 2016), dynamic/conditional inference (Huang et al., 2018; Rao et al., 2021), selective prediction with abstention (Geifman & El-Yaniv, 2017), and capacity-elastic networks (Yu et al., 2019; Cai et al., 2020)—and recent surveys review system-level progress (Zhou et al., 2023; Han et al., 2023). However, practical gaps remain when deploying heterogeneous, post-training cascades under explicit accuracy tolerances, compute/latency budgets, and stability requirements.

A cascaded expert architecture orders multiple pre-trained models from cheap to expensive and routes each input through the chain: if a stage-specific confidence threshold is met, the prediction is accepted; otherwise the input is deferred to the next stage. This black-box composition avoids joint training, naturally handles easy vs. hard examples, and aligns with compute-limited deployments. We therefore adopt a post-training cascade as our foundation and focus on learning stage thresholds and execution order under explicit accuracy tolerance and stability considerations.

Despite their promise, existing cascaded expert systems face several obstacles:

- **Training complexity for long cascades.** Optimizing long chains involves discrete accept/defer decisions and coupled thresholds, yielding a non-differentiable, combinatorial objective (e.g., search over model subset, ordering, and thresholds). With a fixed $K$-stage cascade and $M$ discrete levels per threshold, the threshold search scales as $M^{K-1}$, making exhaustive search infeasible; heuristic or greedy procedures often settle at local optima. End-to-end relaxations (soft routing via Gumbel/straight-through, or policy-gradient methods) suffer from high-variance gradients and brittle credit assignment that worsen with chain length, which is why most prior work restricts to short chains.

- **Objective ambiguity.** Many methods collapse accuracy (or risk) and compute/latency into a single weighted sum. The weights are hard to tune, opaque, and frequently misaligned with concrete requirements such as service-level agreements (e.g., strict p95/p99 latency or per-request cost caps), explicit energy budgets, or an explicit tolerance on accuracy degradation.
- **Stability blind spot.** Stability and generalization vary with design choices, including the chain length $K$, the chosen model set and its ordering, and the thresholds $\tau$, yet these effects are rarely measured or reported.
- **Limited black-box compatibility.** Many pipelines require joint training or architectural changes (e.g., routers/early exits), preventing drop-in use of heterogeneous, independently trained models.

Motivated by these gaps, we propose a stability-aware, post-training cascade for heterogeneous model pools. Under a specified accuracy tolerance relative to the best single model, we freeze all base models and learn only cascade-level controls: which models to include, their execution order, and stage-wise acceptance thresholds that decide accept vs. defer. We quantify inference cost as expected multiply–accumulate operations (MACs) per input which serves as a hardware-agnostic, reproducible proxy correlated with latency and energys.

The pipeline is as follows. We first build a small candidate pool by $\varepsilon$-Pareto screening on the accuracy–compute plane,then build cascades by recursively prepending a light expert and searching stage thresholds with a small local search around the tail. The final cascade and execution order are chosen by a simple stability–compute rank aggregation across cross-validation folds.We validate our method on three datasets: GLUE/SST-2, CIFAR-10, and UrbanSound8K.

## 1.2 Related Work

Efficient inference has evolved from model-local compression to system-aware designs that explicitly trade accuracy for compute at test time. Surveys synthesize this trajectory and position early exiting, dynamic pruning, and cascades as complementary building blocks that modern toolchains can expose and orchestrate (Zhou et al., 2023; Han et al., 2023). Classic cascades and anytime prediction date back to earlier work (e.g., MSDNet, SkipNet, Shallow–Deep, BranchyNet) showing that allocating more compute to harder examples can substantially reduce average cost without large accuracy losses (Huang et al., 2018; Wang et al., 2018; Kaya et al., 2019; Teerapittayanon et al., 2016). In NLP, early-exit BERT variants (DeeBERT, FastBERT, PABEE) adopt confidence- or patience-based halting to accelerate Transformers while controlling degradation (Xin et al., 2020; Liu et al., 2020; Zhou et al., 2020). Our work remains agnostic to the internal mechanisms and instead searches acceptance thresholds over a fixed cascade/order of heterogeneous experts, making it applicable to both classical and neural components.

Cascades and selective routing continue to be effective in deep systems; for instance, Cascade R-CNN demonstrates how quality-aware staging can tighten precision–recall at higher IoU thresholds (Cai & Vasconcelos, 2018). In parallel, capacity-elastic families such as Slimmable/US-Nets and Once-for-All expose compute–accuracy frontiers within a single backbone by width/depth specialization (Yu et al., 2019; Cai et al., 2020). These two lines are orthogonal: elastic backbones reduce per-expert cost, while cascades decide *which* expert to invoke; our method targets the latter by learning thresholds that sit on (or near) the empirical Pareto frontier.

For Transformers and vision models, conditional computation increasingly operates at token, head, or layer granularity. Recent approaches prune tokens or adapt depth online (DynamicViT, EViT, AdaViT), and large-scale sparse Mixture-of-Experts systems push conditional routing to the parameter scale with gated experts (Rao et al., 2021; Liang et al., 2022; Meng et al., 2022; Zoph et al., 2022). These ideas are complementary to our black-box cascade search: dynamic internals can reduce the *per-expert* MACs that enter our objective, while our thresholds determine inter-expert routing under explicit accuracy tolerance.

Beyond mean validation accuracy, stability has been advocated as a selection signal to mitigate overfitting and dataset idiosyncrasies (Hooker, 2021; Pruthi et al., 2022). We instantiate a simple cross-validated stability index and aggregate it with expected MACs via a rank-sum rule (Sec. 2.4), yielding a small set of robust operating points that generalize across folds. This design choice is

pragmatic: it avoids heavyweight Bayesian selection while remaining sensitive to both compute and generalization.

Finally, deployment considerations motivate treating cascades as executable graphs rather than abstract algorithms. Compiler/runtime stacks (e.g., TVM Unity) and system frameworks for conditional/sparse inference (e.g., EdgeMoE) increasingly support dynamic control flow, heterogeneous experts, and hardware-aware scheduling (Chen et al., 2022; Zhang et al., 2023). Our threshold search can thus be slotted into such stacks: once a library of experts is available, the learned thresholds turn into cheap control parameters that gate execution under budget or latency constraints.

## 2 METHODOLOGY

We now describe our methodology. The design of our framework is guided by three goals: to operate over a heterogeneous pool of pre-trained experts without retraining, to explicitly control the trade-off between accuracy and inference cost, and to ensure stability of threshold decisions across folds. These goals motivate the cascade formulation and the search procedures introduced in the following subsections.

### 2.1 CASCADE OF EXPERTS

Let $\{E_1, \ldots, E_M\}$ be experts ordered by increasing MACs. A length-$K$ cascade ($K \leq M$) evaluates experts in this order. Given input $x$, expert $k$ outputs

$$p_k(x) = \mathrm{softmax}(f_k(x)) \in \Delta^{C-1}, \qquad s_k(x) = \max_c [p_k(x)]_c. \tag{1}$$

To ensure only small changes between successive thresholds, we do not threshold raw scores $s_k(x)$ directly. We first round each softmax score to $10^{-3}$ precision and then map it onto a $10^{-2}$-granularity level grid by packing adjacent high-score bins on the training set. Let the descending level set be $\mathcal{L} = \{\ell_1 > \cdots > \ell_L\} = \{0.99, 0.98, \ldots, 0.00\}$. Starting from the top, we merge ("pack") consecutive rounded bins until each mapped level accumulates at least a quota $q$ (we use $q = 1\%$) of training examples; e.g., $\{0.998, 0.997\} \mapsto 0.97$ and $\{0.876, \ldots, 0.872\} \mapsto 0.43$ in Fig. 1. This produces a monotone, many-to-one map $s_k \mapsto \tilde{s}_k$. We take $t_{k,\ell}$ to be the upper boundary for level $\ell$, with $t_{k,\ell_0} := 1$ as a sentinel. The mapped value is obtained by interval lookup:

$$\tilde{s}_k(x) = \ell_i \quad \text{for the unique } i \text{ such that } t_{k,\ell_i} < s_k(x) \leq t_{k,\ell_{i-1}}.$$

We then choose acceptance thresholds $\tau_k \in \mathcal{L}$. The 0.01-step, train-calibrated grid smooths acceptance curves and yields a small, well-behaved search space.

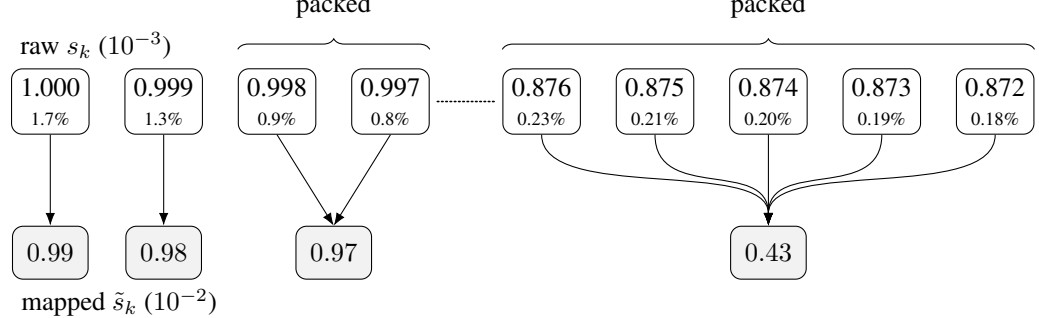

Figure 1: Illustration of the $s_k \rightarrow \tilde{s}_k$ mapping

Each stage $k$ applies a level threshold $\tau_k$:

$$\hat{y}(x) = \begin{cases} \arg\max_c [p_k(x)]_c, & \text{if } \tilde{s}_k(x) \geq \tau_k, \\ \text{forward to } E_{k+1}, & \text{otherwise.} \end{cases}$$

The final expert always accepts ($\tau_K = 0$). On the validation set $V$, let $S_1(\tau) = V$ and define

$$A_k(\tau_k) = \{x \in S_k(\tau) : \tilde{s}_k(x) \geq \tau_k\}, \qquad S_{k+1}(\tau) = S_k(\tau) \setminus A_k(\tau_k).$$

Because the step size is small (0.01) and each threshold bin contains a comparable number of samples, the acceptance curve forms a smooth staircase, making threshold search stable and efficient.

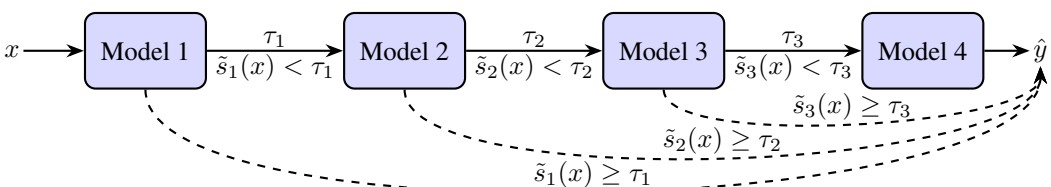

Figure 2: 4-stage cascade. At stage $k$, if $\tilde{s}_k(x) \geq \tau_k$ then exit (accept prediction), otherwise forward to $E_{k+1}$.

## 2.2 PARETO SELECTION WITH ACCURACY TOLERANCE

We describe each model $m \in \mathcal{M}$ by accuracy $\mathrm{ACC}(m)$ and compute $\mathrm{MACs}(m)$. We first form the strict Pareto frontier (with at least one of the two inequalities understood to be strict):

$$\mathcal{P}_0 = \Big\{ m \in \mathcal{M} : \nexists m' \in \mathcal{M} \text{ s.t. } \mathrm{MACs}(m') \leq \mathrm{MACs}(m) \text{ and } \mathrm{ACC}(m') \geq \mathrm{ACC}(m) \Big\}.$$

Since finite-sample noise can create spurious separations on $\mathcal{P}_0$, we refine it with an accuracy tolerance $\epsilon$ and keep only points not $\epsilon$-dominated:

$$\mathcal{P}_\epsilon = \Big\{ m \in \mathcal{P}_0 : \nexists m' \in \mathcal{P}_0 \text{ s.t. } \mathrm{MACs}(m') \leq \mathrm{MACs}(m) \text{ and } \mathrm{ACC}(m') \geq \mathrm{ACC}(m) - \epsilon \Big\}.$$

(Again, in both definitions at least one inequality is strict.)

We set $\epsilon$ to the Wilson half-width of a binomial proportion, treating $\mathrm{ACC}(m)$ as $\hat{p}_m$ from $N$ trials; this captures sampling noise even near 0 or 1, we include a floor $1/N$ to avoid degeneracy, and we use a single worst-case tolerance over $\mathcal{P}_0$ for conservative, monotone pruning that collapses nearties without depending on model-specific $p_m$, so that with 95% confidence accuracy gaps $\leq \epsilon$ are statistically indistinguishable and pruning $m$ as $\epsilon$-dominated by $m'$ implies $m'$ is not truly worse than $m$ by more than $\epsilon$ while using no more compute:

$$\epsilon = \max_{m \in \mathcal{P}_0} \max\left( \frac{1}{N},\ z_{\alpha/2}\sqrt{\frac{p_m(1-p_m)}{N}} \right), \quad p_m = \mathrm{ACC}(m),\ z_{\alpha/2} = 1.96.$$

## 2.3 PREPEND-AND-REUSE THRESHOLD SEARCH WITH LOCAL TAIL REFINEMENT

We impose an upper bound on the deployment accuracy drop compared to the best single model by a user budget $\gamma$. Because validation estimates are noisy (finite $N$), we tighten this budget with a Wilson lower bound while treating $\mathrm{ACC}_{\max}$ as a fixed benchmark to avoid double-counting uncertainty. We then search thresholds $\tau = (\tau_1, \ldots, \tau_K)$ that minimize expected compute subject to

$$\mathrm{ACC}(\tau) \geq \mathrm{ACC}_{\max} - \hat{\gamma}. \tag{2}$$

Let $z = z_{\alpha/2}$ and define the Wilson lower bound for a binomial proportion by

$$\mathrm{LB}(p; N) = \frac{p + \frac{z^2}{2N}}{1 + \frac{z^2}{N}} - \frac{z}{1 + \frac{z^2}{N}} \sqrt{\frac{p(1-p)}{N} + \frac{z^2}{4N^2}}.$$

Set $T_{\mathrm{acc}} = \mathrm{ACC}_{\max} - \gamma$ and

$$p_\star = \min\{ p \in [0,1] :\ \mathrm{LB}(p; N) \geq T_{\mathrm{acc}} \}, \qquad \hat{\gamma} = \mathrm{ACC}_{\max} - p_\star,$$

and write $\delta(\tau) := \mathrm{ACC}_{\max} - \mathrm{ACC}(\tau)$. Equivalently,

$$\hat{p}_\tau \geq p_\star \iff \mathrm{LB}(\hat{p}_\tau; N) \geq \mathrm{ACC}_{\max} - \gamma \iff \delta(\tau) \leq \hat{\gamma}.$$

This tightening controls sampling variability on the validation split under an i.i.d. assumption; it does not correct dataset-level shift between validation and deployment.

**Why prepend-and-reuse works (informal).** Let $c_\star^{(K)} = (\tau_2^\star, \ldots, \tau_K^\star)$ be a $K$-stage tail that meets the tolerance $\hat{\gamma}$ with minimal expected MACs. To form a $(K{+}1)$-stage cascade we prepend a lightweight expert with threshold $\tau_1$ and, starting from $\tau_1{=}1.00$, decrease it in small steps (e.g., $\Delta = 0.01$). For any fixed $\tau_1$, only the harder residual examples are routed to the tail. Because the routed set expands monotonically as $\tau_1$ decreases and the grid is finely spaced, the tail's objective (accuracy under $\hat{\gamma}$ vs. MACs) varies smoothly with respect to $(\tau_2, \ldots, \tau_{K+1})$. Hence the $(K{+}1)$-stage optimum for a given $\tau_1$ typically lies near a good $K$-stage tail. Detailed proofs are deferred to Appendix A.

**First-layer oracle bound (pruning for $\tau_1$).** Since our recursive refinement only decreases the first-layer threshold $\tau_1$, we specialize the pruning rule to $\tau_1$. Let

$$B_{\text{tail}}(x) := \mathbb{1}\{\exists\, \ell > 1 : E_\ell(x) = y\}$$

be the per-sample indicator that some downstream expert could be correct. Then the oracle upper bound on achievable accuracy given $\tau_1$ and a fixed tail configuration is

$$\text{UB}_1(\tau_1, \text{tail}) = \frac{1}{|V|} \sum_{x \in V} \left[ \mathbb{1}\{x \in A_1(\tau_1)\} \cdot \mathbb{1}\{\hat{y}(x) = y\} + \mathbb{1}\{x \notin A_1(\tau_1)\} \cdot B_{\text{tail}}(x) \right]. \quad (3)$$

If $\text{UB}_1(\tau_1, \text{tail}) < \text{ACC}_{\max} - \hat{\gamma}$, the branch cannot meet the tolerance and is pruned. In practice, $B_{\text{tail}}(x)$ is precomputed from stored per-sample correctness of experts $\{E_2, \ldots, E_K\}$ and reused during the $\tau_1$ descent.

$$\tau_{1,\min} := \min \left\{ \tau_1 \in \mathcal{T} : \exists\, \tau_{2:K} \in \mathcal{T}^{K-1} \text{ s.t. } \text{UB}_1(\tau_1, \tau_{2:K}) \geq \text{ACC}_{\max} - \hat{\gamma} \right\}.$$

**From $K$ to $K{+}1$ via prepend-and-reuse** Given a fixed cascade order of experts (sorted by MACs), we grow the depth from $K$ to $K{+}1$ by prepending a first-layer threshold $\tau_1$ to each feasible $K$-level tail $c = (\tau_2, \ldots, \tau_{K+1}) \in \mathcal{F}_{\hat{\gamma}}^{(K)}$. We start at the sentinel $\tau_1{=}1.00$ and iteratively decrease it by a coarse outer step $\Delta$ until either no-improvement persists for $T$ steps or $\tau_1$ reaches a floor $\tau_{1,\min}$. At each $\tau_1$ we perform a local tail refinement around the current tail center $c$ on a Cartesian window of radius $W$ with grid spacing $h$, and pick the feasible point (respecting $\text{ACC} \geq \text{ACC}_{\max} - \hat{\gamma}$) that minimizes expected MACs. The tail center is updated to this best local point; if it improves MACs, the incumbent $(\tau_1, \tau_{2:K+1})^\star$ is refreshed, otherwise a no-improvement counter advances. After the loop for each tail, we add only its best incumbent $(\tau_1, \tau_{2:K+1})^\star$ to $\mathcal{F}_{\hat{\gamma}}^{(K+1)}$. We then return the minimal-MACs solution from the resulting feasible set. See Appendix. B for an illustration of the search procedure and cost comparison.

**Hyperparameter calibration.** To validate the correctness of our recursive procedure and to select suitable hyperparameters $(\Delta, W, h, T)$, we rely on exhaustive cases with small depth. Specifically, we take the optimal 2-stage cascade as the tail and compare (i) a full 3-stage exhaustive search with (ii) the result of recursively prepending from the 2-stage tail. By manually adjusting $(\Delta, W, h, T)$ and repeating this comparison, we tune the parameters until the recursive result matches the exhaustive optimum on $K{=}3$, after which we fix them for larger $K$.

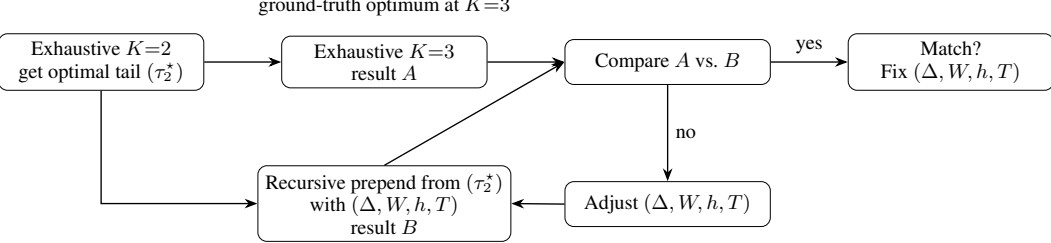

Figure 3: Calibrating $(\Delta, W, h, T)$ by matching the recursive $K \to 3$ result to the exhaustive $K{=}3$ optimum using the optimal 2-stage tail as the seed.

---

**Algorithm 1:** Prepend-and-Reuse: From $K$ to $K+1$ with Local Tail Refinement

---

**Input:** Feasible tails $\mathcal{F}_{\hat{\gamma}}^{(K)}$ (each tail is $(\tau_2, \ldots, \tau_{K+1})$), grid $\mathcal{T} = \{1.00, 0.99, \ldots, 0.00\}$,
tolerance $\hat{\gamma}$, confidence level $\alpha$, step $\Delta$, window $W$, grid $h$, minimal threshold $\tau_{1,\min}$,
no-improve patience $T$

**Output:** Feasible set $\mathcal{F}_{\hat{\gamma}}^{(K+1)}$ and its minimal-MACs solution(s)

$\mathcal{F}_{\hat{\gamma}}^{(K+1)} \leftarrow \emptyset$.

**foreach** *tail* $c = (\tau_2, \ldots, \tau_{K+1}) \in \mathcal{F}_{\hat{\gamma}}^{(K)}$ **do**
   $\tau_1 \leftarrow 1.00$; no_imp_cnt $\leftarrow 0$; $(\tau_1, \tau_{2:K+1})^\star \leftarrow$ None; MACs$^\star \leftarrow +\infty$.
   **repeat**
      `// Local Cartesian grid around current tail center c: ±W`
         `with step h`
      $\mathcal{N} \leftarrow \{\tau'_{2:K+1} \in \mathcal{T}^K : \|\tau'_{2:K+1} - c\|_\infty \leq W, \text{ step } h\}$
      `// Pick feasible point with minimal expected MACs under`
         `tolerance`
      $\tau_{2:K+1}^{\text{new}} \leftarrow \arg\min_{\tau'_{2:K+1} \in \mathcal{N}} \text{MACs}(\tau_1, \tau'_{2:K+1}) \text{ s.t. } \text{ACC}(\tau_1, \tau'_{2:K+1}) \geq \text{ACC}_{\max} - \hat{\gamma}$.
      **if** $\tau_{2:K+1}^{\text{new}}$ *exists* **then**
         $c \leftarrow \tau_{2:K+1}^{\text{new}}$                      `// tail center update`
         **if** $\text{MACs}(\tau_1, c) < \textit{MACs}^\star$ **then**
            $(\tau_1, \tau_{2:K+1})^\star \leftarrow (\tau_1, c); \quad \text{MACs}^\star \leftarrow \text{MACs}(\tau_1, c); \quad$ no_imp_cnt $\leftarrow 0$
         **else**
            no_imp_cnt $\leftarrow$ no_imp_cnt $+1$
      $\tau_1 \leftarrow \max\{\tau_1 - \Delta, \ \tau_{1,\min}\}$
   **until** *no_imp_cnt* $= T$ **or** $\tau_1 \leq \tau_{1,\min}$
   `// Add only the best candidate found for this tail`
   **if** $(\tau_1, \tau_{2:K+1})^\star$ *exists* **then**
      $\mathcal{F}_{\hat{\gamma}}^{(K+1)} \leftarrow \mathcal{F}_{\hat{\gamma}}^{(K+1)} \cup \{(\tau_1, \tau_{2:K+1})^\star\}$

**return** minimal-MACs solution(s) from the filtered $\mathcal{F}_{\hat{\gamma}}^{(K+1)}$.

---

**Degenerate cascade filtering (all depths).** After collecting all learned cascades across depths, we remove degenerate solutions whose thresholds hit the boundaries. Concretely, for any cascade of depth $K$, if there exists a stage $j \in \{1, \ldots, K\}$ with $\tau_j \in \{0.00, 1.00\}$, we discard the cascade. A threshold $\tau_j = 1.00$ means the stage never accepts, so the stage effectively disappears; a threshold $\tau_j = 0.00$ means the stage always accepts, so it nullifies the intended gating. We therefore keep only cascades with $\tau_j \in (0, 1)$ for all stages.

## 2.4 CASCADE & ORDER SELECTION VIA STABILITY–COMPUTE RANK AGGREGATION

To assess generalization of thresholds, we use $K_{\text{cv}}$-fold cross-validation with equal-sized, stratified splits. For each fold $i$, thresholds $\tau$ are re-optimized from scratch on the training split using the same search hyperparameters and tolerance, and then evaluated once on the corresponding validation split. We define

$$\text{StabIndex} = \frac{1}{K_{\text{cv}}} \sum_{i=1}^{K_{\text{cv}}} \left| \text{ACC}_{\text{train}}^{(i)} - \text{ACC}_{\text{val}}^{(i)} \right|. \tag{4}$$

A smaller $\text{StabIndex}$ indicates better generalization of the learned thresholds.

We note there are many reasonable ways to select a cascade from stability and compute. In this work we adopt a simple rank-sum rule: for each candidate $c$ compute $\text{StabIndex}(c)$ and expected compute $\text{MACs}(c)$, let $r_{\text{stab}}(c)$ and $r_{\text{mac}}(c)$ be their ascending ranks within $\mathcal{C}$, and choose

$$c^\star = \arg\min_{c \in \mathcal{C}} \left[ r_{\text{stab}}(c) + r_{\text{mac}}(c) \right].$$

## 3 EXPERIMENTS

### 3.1 DATASETS AND EXPERIMENTAL SETUP

To assess cross-modal generality, we evaluate on three datasets—text (GLUE / SST-2), audio (UrbanSound8K), and vision (CIFAR-10). For each dataset, we construct a cascade from a pool of base models. The model list and configurations are in Appendix C, and the model-pools (MACs, validation accuracy, and test accuracy) are in Appendix D[1].

**Dataset A: GLUE / SST-2.** Binary sentence-level sentiment classification on English text (2 classes; 70,042 total examples). We use the standard GLUE split: 67,349 train, 872 validation, and 1,821 test. The model pool contains 6 architectures (TFIDF–LR, W2V–DAN, RNN–BiGRU, TextCNN, SBERT–GCN, BERT), yielding 116 base models.

**Dataset B: CIFAR-10.** Image classification with 10 classes and 60,000 images at $32 \times 32$ resolution. We use 45,000 train, 5,000 validation, and 10,000 test images. The pool contains 10 architectures (HOG–SVM, SimpleCNN, MLP, ResNet, MobileNetV2, ShuffleNetV2, GhostNet, DeiT, Swin, ConvNeXt), totaling 51 base models.

**Dataset C: UrbanSound8K.** Environmental sound classification with 10 classes and 8,732 clips, partitioned into 10 official folds. We follow a fixed protocol: folds 1–7 for training, folds 8–9 for validation, and fold 10 for testing, giving 6,273 / 1,622 / 837 samples, respectively. The pool contains 6 architectures (TinyCNN, CRNN, ResNet, PANN, AST, LSTM), totaling 49 base models.

Table 1: Datasets and model pools.

| Dataset | #Classes | Total Samples | Split (Train / Val / Test) | Pool (archs) | Base models |
|---|---|---|---|---|---|
| GLUE / SST-2 | 2 | 70,042 | 67,349 / 872 / 1,821 | 6 | 116 |
| CIFAR-10 | 10 | 60,000 | 45,000 / 5,000 / 10,000 | 10 | 51 |
| UrbanSound8K | 10 | 8,732 | 6,273 / 1,622 / 837 | 6 | 49 |

All experiments use PyTorch with CUDA and AMP, primarily on a single NVIDIA RTX 4090 (24 GB); MACs are reported per input and exclude non-learned preprocessing.

### 3.2 PARETO FRONTIER RESULTS

We compute pointwise-$\varepsilon$ Pareto frontiers for each dataset (over all base models). All frontier plots are deferred to Appendix D.2.

For downstream analysis, we select a small set of operating points from the $\varepsilon$-Pareto frontier on each dataset:

- **GLUE/SST-2:** 6 frontier points (TFIDF-LR-1, TFIDF-LR-3, TextCNN-1, RNN-BiGRU-24, BERT-4, BERT-6).
- **CIFAR-10:** 7 frontier points (HOG–SVM–3, MLP-2, SimpleCNN-1, SimpleCNN-2, GhostNet-1, GhostNet-2, GhostNet-6).
- **UrbanSound8K:** 4 frontier points (LSTM-7, CRNN-3, TinyCNN-5, CRNN-8).

### 3.3 TRAINING COMPLEXITY REDUCTION

We compare the number of calls (Appendix. B) required by exhaustive search and by our recursive scheme. Across GLUE/SST-2, CIFAR-10, and UrbanSound8K and tolerances $\gamma \in \{0.03, 0.05, 0.08\}$, our method requires far fewer calls, yielding orders-of-magnitude reductions.

---

[1]For GLUE/SST-2, single-model test accuracy is not reported due to the hidden test set and submission limits.

We adopt the same search hyperparameters throughout: outer step $\Delta = 0.01$, local window radius $W = 0.06$, inner grid step $h = 0.01$, and no-improve patience $T = 8$.

Table 2: Calls vs. accuracy tolerance $\gamma$ (lower is better). Exhaustive counts are independent of $\gamma$.

| Dataset | Method | $\gamma$=0.03 | $\gamma$=0.05 | $\gamma$=0.08 |
|---------|--------|---------------|---------------|---------------|
| GLUE/SST-2 | Exhaustive | 63,060,603,000 | 63,060,603,000 | 63,060,603,000 |
| | **Ours** | 10,916,265 | 14,193,655 | 20,575,315 |
| CIFAR-10 | Exhaustive | 7,430,641,054,200 | 7,430,641,054,200 | 7,430,641,054,200 |
| | **Ours** | 76,765,825 | 54,306,341 | 49,107,272 |
| UrbanSound8K | Exhaustive | 4,121,200 | 4,121,200 | 4,121,200 |
| | **Ours** | 37,270 | 34,614 | 28,936 |

**Effect of cascade length $k$.** From Appendix. B, exhaustive scales as

$$N_{\text{exh}}^{\text{calls}} = \sum_{k=2}^{K} \binom{M_{\text{pool}}}{k} k\, M^{k-1},$$

which is combinatorial in $M_{\text{pool}}$ and grows rapidly with $k$ via $M^{k-1}$. Our recursive search satisfies

$$N_{\text{ours}}^{\text{calls}} \lesssim \sum_{k=2}^{K} H_{\max}(k)\, |\mathcal{F}_{\hat{\gamma}}^{(k-1)}|\, k\, S\, R^{k-2},$$

with $H_{\max}(k)$, $S$, and $R$ defined in Appendix. B. Our method makes two substitutions: (i) the exponential threshold grid $M^{k-1}$ is replaced by constant-size scans/windows $(S, R)$; and (ii) the combinatorial head–tail count $k\binom{M_{\text{pool}}}{k}$ is replaced by $H_{\max}(k)\, |\mathcal{F}_{\hat{\gamma}}^{(k-1)}|$ with $H_{\max}(k)\, |\mathcal{F}_{\hat{\gamma}}^{(k-1)}| \ll k\binom{M_{\text{pool}}}{k}$. Consequently $N_{\text{ours}}^{\text{calls}} \ll N_{\text{exh}}^{\text{calls}}$, and the advantage grows with $k$.

**Effect of accuracy tolerance $\gamma$.** The dependence on $\gamma$ can be non-monotonic: larger $\gamma$ enlarges the feasible region ($\uparrow$ more tails/branches to expand), but also enables earlier acceptance and stronger pruning ($\downarrow$ shorter scans). Empirically (Table 2), SST-2 shows a slight increase with $\gamma$, and CIFAR-10,UrbanSound8K decreases; exhaustive is unaffected by $\gamma$.

### 3.4 STABILITY AND MACs ANALYSIS

We analyze the Stability Index $\text{StabIndex}$ across folds (definition in Sec. 2.4). Across three datasets (GLUE/SST-2, CIFAR-10, and UrbanSound8K) and tolerances $\gamma \in \{0.03, 0.05, 0.08\}$, we study the trade-off between stability and expected MACs. All panels for all (dataset, $\gamma$) combinations are deferred to Appendix E.2. We use these results to identify the optimal cascade model and its execution order for each dataset and tolerance (full tables in Appendix E.1). Unless otherwise noted, all results use 5-fold stratified cross-validation with identical splits across methods.

### 3.5 FINAL RESULTS UNDER ACCURACY TOLERANCE

We evaluate cascades under accuracy tolerances $\gamma$. On GLUE/SST-2 and CIFAR-10, our cascades satisfy the $\gamma$-tolerance while delivering substantial MACs reductions. On UrbanSound8K, the test accuracy drop exceeds $\gamma$; we regard this as a strict held-out test effect rather than a limitation of the cascade itself, given the much larger validation–test mismatch (mean $|\text{Test} - \text{Val}| = 0.043647$ on UrbanSound8K vs. $0.004122$ on CIFAR-10). Moreover, cross-validation stability remains small across modalities and tolerances (GLUE/SST-2: 0.0165/0.0123/0.0145; CIFAR-10: 0.00545/0.00385/0.00715; UrbanSound8K: 0.0187/0.0140/0.0140 for $\gamma$=0.03/0.05/0.08), indicating that our cascade selection—especially on Audio—was stable rather than misjudged. Since our method assembles cascades from fixed, pre-trained base models without retraining or domain adaptation, it is not designed to correct dataset-level distribution shifts; the deviation on UrbanSound8K is therefore consistent with this mismatch rather than with the cascade mechanism.

Table 3: Final cascades selected per dataset and tolerance $\gamma$.

| Dataset | $\gamma$ | $K$ | Model(s) | Thresholds $\tau$ |
|---|---|---|---|---|
| GLUE / SST-2 | 0.03 | 3 | TFIDF-LR-3 $\rightarrow$ BERT-4 $\rightarrow$ BERT-6 | [0.89 0.89] |
| GLUE / SST-2 | 0.05 | 4 | TFIDF-LR-1 $\rightarrow$ TFIDF-LR-3 $\rightarrow$ BERT-4 $\rightarrow$ BERT-6 | [0.84 0.79 0.83] |
| GLUE / SST-2 | 0.08 | 4 | TFIDF-LR-1 $\rightarrow$ TFIDF-LR-3 $\rightarrow$ RNN-BiGRU-24 $\rightarrow$ BERT-4 | [0.89 0.89 0.97] |
| CIFAR-10 | 0.03 | 2 | SimpleCNN-1 $\rightarrow$ GhostNet-2 | [0.57] |
| CIFAR-10 | 0.05 | 2 | SimpleCNN-1 $\rightarrow$ GhostNet-1 | [0.49] |
| CIFAR-10 | 0.08 | 2 | SimpleCNN-1 $\rightarrow$ GhostNet-1 | [0.35] |
| UrbanSound8K | 0.03 | 2 | TinyCNN-5 $\rightarrow$ CRNN-8 | [0.35] |
| UrbanSound8K | 0.05 | 2 | CRNN-3 $\rightarrow$ CRNN-8 | [0.32] |
| UrbanSound8K | 0.08 | 2 | CRNN-3 $\rightarrow$ TinyCNN-5 | [0.23] |

Table 4: Final cascade vs. best single model under different accuracy tolerances $\gamma$.

| Dataset | $\gamma$ | Cascade | | Best Single | | MACs$\downarrow$ (%) | Acc Drop |
|---|---|---|---|---|---|---|---|
| | | Acc | MACs | Acc | MACs | | |
| GLUE / SST-2 | 0.03 | 0.943 | 1.16e9 | 0.964 | 4.93e9 | 76% | 0.021 |
| GLUE / SST-2 | 0.05 | 0.925 | 0.80e9 | 0.964 | 4.93e9 | 84% | 0.039 |
| GLUE / SST-2 | 0.08 | 0.900 | 0.23e9 | 0.964 | 4.93e9 | 95% | 0.064 |
| CIFAR-10 | 0.03 | 0.900 | 0.47e8 | 0.919 | 4.22e8 | 89% | 0.019 |
| CIFAR-10 | 0.05 | 0.879 | 0.18e8 | 0.919 | 4.22e8 | 96% | 0.040 |
| CIFAR-10 | 0.08 | 0.850 | 0.12e8 | 0.919 | 4.22e8 | 97% | 0.069 |
| UrbanSound8K | 0.03 | 0.769 | 0.63e9 | 0.809 | 1.21e9 | 48% | 0.040 |
| UrbanSound8K | 0.05 | 0.706 | 0.48e9 | 0.809 | 1.21e9 | 60% | 0.103 |
| UrbanSound8K | 0.08 | 0.658 | 0.23e9 | 0.809 | 1.21e9 | 81% | 0.151 |

## 4 CONCLUSION

We presented a stability-aware, post-training cascade of experts that operates over heterogeneous pretrained models and learns only cascade-level controls—execution order and stage thresholds—under an explicit accuracy tolerance. The method first forms a compact candidate pool via $\varepsilon$-Pareto screening on the accuracy–compute plane, then grows cascades with a prepend-and-reuse threshold search that refines tails on a small mapped-confidence grid, and finally selects the deployed cascade by aggregating cross-validated stability with expected MACs. Across text (GLUE/SST-2), vision (CIFAR-10), and audio (UrbanSound8K), the approach achieves large reductions in expected compute while adhering to the user-set tolerance when the validation–test gap is not large, and it cuts threshold-search cost by orders of magnitude compared to exhaustive search.

## USE OF LLMS

During the preparation of this paper, we used OpenAI GPT-5 to assist with language polishing and formatting. All conceptual contributions, experimental designs, and conclusions were developed by the authors, who take full responsibility for the content of this paper.

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

# Appendix

APPENDIX CONTENTS

## A   THEORETICAL GUARANTEES FOR RECURSIVE THRESHOLD SEARCH

**Problem and notation.**   Data $(x, y) \sim \mathcal{D}$. There are $K$ experts $E_1, \ldots, E_K$ ordered by non-decreasing per-sample cost $0 < \kappa_1 \leq \cdots \leq \kappa_K$ (MACs). Expert $i$ outputs a softmax $p_i(x)$; let $s_i(x) = \max_j p_i^{(j)}(x)$ be the raw max-probability. To stabilize thresholding we apply a *monotone packing/quantization*

$$\tilde{s}_i(x) = \Phi_i(s_i(x)), \qquad \Phi_i : [0, 1] \to \mathcal{L},$$

where the threshold grid follows the mapped-confidence definition in Sec. 2.1:

$$\mathcal{T} = \{\, 1.00, \ 0.99, \ 0.98, \ \ldots \}.$$

Let $\tau = (\tau_1, \ldots, \tau_K) \in \mathcal{L}^K$. Stage $i$ *accepts* iff $\tilde{s}_i(x) \geq \tau_i$, otherwise the example is deferred to $i{+}1$. Let the stopping stage be $I_\tau(x) \in \{1, \ldots, K\}$ and the acceptance set $A_i(\tau) = \{x : I_\tau(x) = i\}$. Expected compute and population accuracy are

$$C(\tau) = \mathbb{E}_x\Big[ \sum_{j=1}^{I_\tau(x)} \kappa_j \Big], \qquad \mathrm{ACC}(\tau) = \mathbb{P}_{(x,y)}\big(\hat{y}_{I_\tau(x)}(x) = y\big).$$

Let $\mathrm{ACC}_{\max} = \max_i \mathbb{P}(\hat{y}_i(x) = y)$ be the best single-expert accuracy and $\gamma \in [0, 1]$ a tolerance. We optimize in the *discrete domain*

$$\min_{\tau \in \mathcal{L}^K} C(\tau) \qquad \text{s.t.} \qquad \mathrm{ACC}(\tau) \ \geq \ \mathrm{ACC}_{\max} - \gamma. \tag{5}$$

**Assumption 1 (structural).**

**Assumption 1** (A1: Monotone success / stochastic dominance)**.**  For any $x$, later experts are no worse: $p_1(x) \leq p_2(x) \leq \cdots \leq p_K(x)$ where $p_i(x) := \mathbb{P}(\hat{y}_i(x) = y \mid x)$.

**Mild regularity (used only for locality).**

**Assumption 2** (R1: Bounded local mass under the mapped-coverage grid)**.**  For each stage $i$, in a neighborhood of $\tau_i$ the marginal of the mapped score $\tilde{s}_i$ has bounded mass per grid bin. Equivalently, on a continuous scale the density is bounded by $f_i^\star < \infty$; on a grid with step size $\Delta \in (0, 1]$, each bin carries at most $f_i^\star \Delta$ probability mass.

We work throughout on a mapped-coverage (quantile) grid as defined in Sec. 2.1, i.e., thresholds are placed on approximately equal-spaced quantiles of the score distribution. Consequently, at the population level two adjacent grid points differ by roughly $\Delta$ in cumulative mass (e.g., $\Delta{=}0.01$ yields an "about 1% per step" change), while in finite samples the deviation is small and vanishes with data size. This construction avoids pathological spikes or plateaus in raw confidence scores—situations that could violate bounded per-bin mass—and makes the assumption above both natural and practically verifiable.

**Assumption 3** (R2: Local Lipschitz of accuracy in coverage)**.**  Let $\mathrm{cov}_i(\tau_i) = \mathbb{P}(\tilde{s}_i(x) \geq \tau_i \mid x \text{ reaches } i)$. There exists $L_{\mathrm{ACC},i} > 0$ such that for small perturbations

$$\big|\Delta\mathrm{ACC}\big| \ \leq \ L_{\mathrm{ACC},i} \, \big|\Delta\mathrm{cov}_i\big|.$$

**L1 (Monotone reparametrization invariance).**

**Lemma 1** (L1: Invariance)**.**  If $\tilde{s}_i = \Phi_i(s_i)$ with $\Phi_i$ non-decreasing, then for any $t \in \mathcal{L}$, $\{x : \tilde{s}_i(x) \geq t\} = \{x : s_i(x) \geq t\}$. Hence any threshold property proved in the $\tilde{s}$-space holds identically in the raw $s$-space.

*Proof.* $\tilde{s}_i(x) \geq t \Leftrightarrow \Phi_i(s_i(x)) \geq \Phi_i(t) \Leftrightarrow s_i(x) \geq t$ by monotonicity. $\qquad\square$

**L2 (Directional effect of thresholds).**

**Lemma 2** (L2: Threshold directionality)**.**  Fix $\tau$ and lower a single threshold to $\tau_k' < \tau_k$ (earlier exits are easier). Then

$$C(\tau') \leq C(\tau), \qquad \mathrm{ACC}(\tau') \leq \mathrm{ACC}(\tau).$$

*Sketch.* Lowering $\tau_k$ enlarges $A_k$ so more samples exit earlier $\Rightarrow$ fewer deep computations $\Rightarrow$ $C$ non-increasing. Under A1, earlier exits include more borderline examples, so overall accuracy is non-increasing. $\qquad\square$

**L3 (Boundary optimality in the discrete domain).**

**Lemma 3** (L3: Boundary optimality). *If $\tau^\star$ solves equation 5, then for any coordinate $i$ that can be further lowered by one bin, $\mathrm{ACC}(\tau^\star - \Delta e_i) < \mathrm{ACC}_{\max} - \gamma$. Equivalently, $\tau^\star$ lies on the discrete equality boundary.*

*Proof.* If $\tau^\star$ had slack, L2 allows lowering some coordinate to strictly reduce $C$ while staying feasible, contradicting optimality. $\qquad\square$

**L4 (Bounded drift / locality).**

**Lemma 4** (L4: Locality). *Lower only the first threshold by $\delta > 0$. Then*

$$|\Delta \mathrm{cov}_1| \leq f_1^\star \delta, \qquad |\Delta \mathrm{ACC}| \leq L_{\mathrm{ACC},1} f_1^\star \delta.$$

*Hence there exists a tail adjustment $\tau'_{2:K}$ with $\|\tau'_{2:K} - \tau_{2:K}\|_\infty \leq K_{\mathrm{loc}}\delta$ that restores the equality boundary of L3.*

*Sketch.* R1 yields the coverage change bound; composing with R2 gives the accuracy change bound. Compensating this change with tail thresholds requires only $O(\delta)$ movement. $\qquad\square$

**L5 (Hausdorff bound for boundary sets & neighborhood of the new optimum).** Define the tail sets on the equality boundary before/after the outer step:

$$\mathcal{B}(\tau_1) = \{\mathbf{u} \in \mathcal{L}^{K-1} : \mathrm{ACC}(\tau_1, \mathbf{u}) = \mathrm{ACC}_{\max} - \gamma\},$$

$$\mathcal{B}(\tau_1 - \delta) = \{\mathbf{u} \in \mathcal{L}^{K-1} : \mathrm{ACC}(\tau_1 - \delta, \mathbf{u}) = \mathrm{ACC}_{\max} - \gamma\}.$$

Assume a discrete *transversality* lower bound: there exists $\underline{\sigma} > 0$ such that for any $\mathbf{u}$ on the boundary, moving any tail coordinate $u_i$ by one grid step $\Delta$ changes accuracy by at least $\underline{\sigma}\Delta$ in magnitude.

**Lemma 5** (L5: Hausdorff bound and neighborhood of the new optimum). *Under A1, R1, R2 and the above transversality, the boundary sets satisfy*

$$d_{\mathrm{H}}\big(\mathcal{B}(\tau_1), \mathcal{B}(\tau_1 - \delta)\big) \;\leq\; r(\delta) := \Big\lceil \frac{L_{\mathrm{ACC},1} f_1^\star}{\underline{\sigma}} \cdot \frac{\delta}{\Delta} \Big\rceil \Delta = O(\delta).$$

*Let $\mathbf{u}^\star(\tau_1 - \delta) \in \arg\min_{\mathbf{u} \in \mathcal{B}(\tau_1 - \delta)} C(\tau_1 - \delta, \mathbf{u})$. Then $\mathbf{u}^\star(\tau_1 - \delta) \in \mathcal{N}_{r(\delta)}\big(\mathcal{B}(\tau_1)\big)$, i.e., the new boundary minimizer lies within an $r(\delta)$-thickened neighborhood of the old boundary.*

*Sketch.* By L4, changing $\tau_1$ by $\delta$ alters boundary accuracy by at most $L_{\mathrm{ACC},1} f_1^\star \delta$. Since each tail grid move alters accuracy by at least $\underline{\sigma}\Delta$, at most $\lceil (L_{\mathrm{ACC},1} f_1^\star \delta)/(\underline{\sigma}\Delta) \rceil$ steps suffice to re-attain the boundary. This yields the Hausdorff bound and the neighborhood claim. $\qquad\square$

**Strengthened pointwise closeness.** Assume *local uniqueness/sharpness* of $C$ on the boundary near the old minimizer $\mathbf{u}^\star(\tau_1)$: there exist $r > 0$ and $\mu > 0$ such that for all $\mathbf{u} \in \mathcal{B}(\tau_1) \cap \mathcal{N}_r(\mathbf{u}^\star(\tau_1))$,

$$C(\tau_1, \mathbf{u}) - C(\tau_1, \mathbf{u}^\star(\tau_1)) \;\geq\; \mu \|\mathbf{u} - \mathbf{u}^\star(\tau_1)\|_\infty.$$

Assume also a *local Lipschitz-in-$\tau$* bound, uniform over $\mathbf{u}$ in the same neighborhood:

$$\sup_{\mathbf{u} \in \mathcal{N}_r(\mathbf{u}^\star(\tau_1))} \big| C(\tau_1 - \delta, \mathbf{u}) - C(\tau_1, \mathbf{u}) \big| \;\leq\; L_C \, \delta.$$

Then any new boundary minimizer $\mathbf{u}^\star(\tau_1 - \delta) \in \arg\min_{\mathbf{u} \in \mathcal{B}(\tau_1 - \delta)} C(\tau_1 - \delta, \mathbf{u})$ satisfies

$$\big\|\mathbf{u}^\star(\tau_1 - \delta) - \mathbf{u}^\star(\tau_1)\big\|_\infty \;\leq\; \frac{L_C}{\mu} \delta \,+\, O(\Delta).$$

*However*, such sharpness/uniqueness may fail in practice (e.g., multiple nearly-flat minima along the boundary), so pointwise closeness cannot be guaranteed in general; the set-wise neighborhood of Lemma 5 still holds.

**Engineering mitigation.** To reduce the risk of missing a new optimum when sharpness/uniqueness fails, *modestly enlarge the search neighborhood* around the old boundary in the next step (e.g., expand the tail search radius by a small constant factor for one iteration).

**L6 (Stopping reliability).**

**Lemma 6** (L6: Missed-improvement probability). Assume a coverage/solver success rate $\rho \in (0, 1]$ such that whenever a feasible improvement exists in the current neighborhood, the inner search finds a strictly smaller $C$ with probability at least $\rho$, independently across outer steps (conditionally). With the rule "stop after $L$ consecutive steps with no improvement,"

$$\mathbb{P}(\text{stop while an improvement still exists}) \ \leq \ (1 - \rho)^L.$$

Hence choosing $L \geq \log(\alpha)/\log(1 - \rho)$ controls the miss probability by $\alpha$.

**Main statement**     Under A1, R1, R2 and L5's transversality, consider the recursive search that decreases $\tau_1$ by an outer step $\delta$ and searches only within an $\ell_\infty$ tail neighborhood of radius $r(\delta) = \Theta(\delta)$. At every accepted iterate the solution lies on the equality boundary (L3). Whenever an improvement is found, the compute strictly decreases (L2). If the strengthened sharpness condition holds, the new global boundary minimizer lies within $O(\delta)$ of the previous one (up to grid error $O(\Delta)$); otherwise, the set-wise neighborhood of L5 still ensures that a modestly enlarged neighborhood suffices to capture the new optimum in practice. Finally, the probability of premature stopping is bounded by $(1 - \rho)^L$ (L6).

## B  SEARCH COST VS. EXHAUSTIVE

Let $\mathcal{P}$ be the expert pool with $|\mathcal{P}| = M_{\text{pool}}$ experts, and let $\mathcal{T}$ be the per–expert threshold grid with $M := |\mathcal{T}|$ levels. We write $S := 1 + \lceil (1 - \tau_{1,\min})/\Delta \rceil$ for the 1D scan budget along a new head (step size $\Delta$ from the minimal admissible threshold $\tau_{1,\min}$), and $R := 2W/h + 1$ for the per–stage local window width (radius $W$ on a grid with stride $h$). For $k \geq 2$, let

$$C_k := \big| \{ \{ E_{i_1}, \ldots, E_{i_k} \} \subset \mathcal{P} \} \big| = \binom{M_{\text{pool}}}{k}.$$

We measure search cost by the number of calls, where one call equals running one expert once on the validation set during threshold search.

**Exhaustive search (hypothetical baseline).** Under the common coordinate probing view—each configuration differs from the current one by changing one coordinate at a time—the call complexity for a $k$–tuple scales like $k M^{k-1}$. Summing over all $k$ we get

$$N_{\text{exh}}^{\text{calls}} = \sum_{k=2}^{K} C_k \, k \, M^{k-1}.$$

(If one counts full configurations, the factor becomes $M^k$ instead of $k M^{k-1}$.)

**Recursive search (ours): $K{=}2$ exhaustive seed, $K{\geq}3$ recursive.** Let $\mathcal{F}_{\hat{\gamma}}^{(k-1)}$ denote the set of $(k-1)$–stage feasible tails that survive tolerance $\hat{\gamma}$ (monotone acceptance and $\varepsilon$–screening). For each tail, the number of admissible heads to prepend is at most

$$H_{\max}(k) \leq M_{\text{pool}} - (k - 1) \leq M_{\text{pool}} \quad \text{(safe upper bound).}$$

Expanding one head costs at most $k S R^{k-2}$ calls (one 1D scan of length $S$ for the new head, with a local window of width $R$ per of the remaining $k{-}2$ coordinates), hence with only $k{=}2$ exhaustive and all $k{\geq}3$ recursive:

$$N_{\text{ours}}^{\text{calls}} \leq \underbrace{C_2 \cdot 2M}_{K{=}2 \text{ exhaustive seed}} + \sum_{k=3}^{K} H_{\max}(k) \, \big| \mathcal{F}_{\hat{\gamma}}^{(k-1)} \big| \, k \, S \, R^{k-2}.$$

(The $k{=}2$ case reduces to a simple coordinate scan, hence the explicit first term above.)

**Comparison and takeaway.** Exhaustive grows combinatorially in both the pool size and the threshold grid:

$$N_{\text{exh}}^{\text{calls}} = \Theta\Big( \sum_{k=2}^{K} \binom{M_{\text{pool}}}{k} k \, M^{k-1} \Big) \quad \Big( \text{or } \Theta\big( \binom{M_{\text{pool}}}{k} M^k \big) \text{ for full configs} \Big).$$

Our recursion replaces $M^{k-1}$ by a product of problem-dependent factors $S R^{k-2}$ and, crucially, by the pruned frontier size $|\mathcal{F}_{\hat{\gamma}}^{(k-1)}| \ll \binom{M_{\text{pool}}}{k-1}$:

$$N_{\text{ours}}^{\text{calls}} = \mathcal{O}\Big( \sum_{k=2}^{K} \underbrace{|\mathcal{F}_{\hat{\gamma}}^{(k-1)}|}_{\ll \binom{M_{\text{pool}}}{k-1}} \cdot k \cdot S \cdot R^{k-2} \Big).$$

When the acceptance predicate is monotone and the feasible region is locally "thin" (frontier of measure zero in the grid limit), one obtains $|\mathcal{F}_{\hat{\gamma}}^{(k-1)}| = \tilde{\mathcal{O}}(R^{k-2})$, so $N_{\text{ours}}^{\text{calls}}$ is polynomial in $R$ and $S$ rather than exponential in $M$, matching the multi–order-of-magnitude savings we observe empirically.

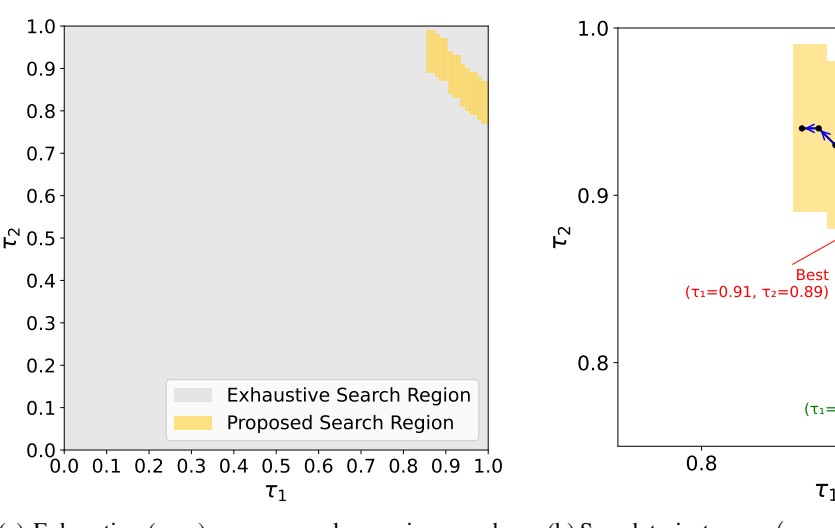

(a) Exhaustive (gray) vs. proposed recursive search region (yellow).

(b) Search trajectory on $(\tau_1, \tau_2)$ with $\pm W$ refinement windows ($\Delta = 0.01$, $W = 0.05$, $T = 5$).

Figure 4: Illustration of the recursive threshold search seeded by $K{=}2$ exhaustive.

# C  MODEL LIST AND CONFIGURATIONS

Table 5: Models used on the GLUE / SST-2 dataset.

| Dataset | Model | Description |
|---|---|---|
| GLUE / SST-2 | TFIDF–LR–1 | word TF–IDF (1–1), V=2k; Logistic Regression. |
| GLUE / SST-2 | TFIDF–LR–2 | word TF–IDF (1–1), V=4k; Logistic Regression. |
| GLUE / SST-2 | TFIDF–LR–3 | word TF–IDF (1–1), V=6k; Logistic Regression. |
| GLUE / SST-2 | TFIDF–LR–4 | word TF–IDF (1–1), V=8k; Logistic Regression. |
| GLUE / SST-2 | TFIDF–LR–5 | word TF–IDF (1–2), V=2k; Logistic Regression. |
| GLUE / SST-2 | TFIDF–LR–6 | word TF–IDF (1–2), V=4k; Logistic Regression. |
| GLUE / SST-2 | TFIDF–LR–7 | word TF–IDF (1–2), V=6k; Logistic Regression. |
| GLUE / SST-2 | TFIDF–LR–8 | word TF–IDF (1–2), V=8k; Logistic Regression. |
| GLUE / SST-2 | TFIDF–LR–9 | word TF–IDF (1–3), V=2k; Logistic Regression. |
| GLUE / SST-2 | TFIDF–LR–10 | word TF–IDF (1–3), V=4k; Logistic Regression. |
| GLUE / SST-2 | TFIDF–LR–11 | word TF–IDF (1–3), V=6k; Logistic Regression. |
| GLUE / SST-2 | TFIDF–LR–12 | word TF–IDF (1–3), V=8k; Logistic Regression. |
| GLUE / SST-2 | W2V–DAN–1 | Word2Vec CBOW, d=100, w=5 → W2V–DAN (h=256). |
| GLUE / SST-2 | W2V–DAN–2 | Word2Vec CBOW, d=100, w=8 → W2V–DAN (h=256). |
| GLUE / SST-2 | W2V–DAN–3 | Word2Vec CBOW, d=100, w=10 → W2V–DAN (h=256). |
| GLUE / SST-2 | W2V–DAN–4 | Word2Vec CBOW, d=200, w=5 → W2V–DAN (h=256). |
| GLUE / SST-2 | W2V–DAN–5 | Word2Vec CBOW, d=200, w=8 → W2V–DAN (h=256). |
| GLUE / SST-2 | W2V–DAN–6 | Word2Vec CBOW, d=200, w=10 → W2V–DAN (h=256). |
| GLUE / SST-2 | W2V–DAN–7 | Word2Vec CBOW, d=300, w=5 → W2V–DAN (h=256). |
| GLUE / SST-2 | W2V–DAN–8 | Word2Vec CBOW, d=300, w=8 → W2V–DAN (h=256). |
| GLUE / SST-2 | W2V–DAN–9 | Word2Vec CBOW, d=300, w=10 → W2V–DAN (h=256). |
| GLUE / SST-2 | W2V–DAN–10 | Word2Vec Skip-gram, d=100, w=5 → W2V–DAN (h=256). |
| GLUE / SST-2 | W2V–DAN–11 | Word2Vec Skip-gram, d=100, w=8 → W2V–DAN (h=256). |
| GLUE / SST-2 | W2V–DAN–12 | Word2Vec Skip-gram, d=100, w=10 → W2V–DAN (h=256). |
| GLUE / SST-2 | W2V–DAN–13 | Word2Vec Skip-gram, d=200, w=5 → W2V–DAN (h=256). |
| GLUE / SST-2 | W2V–DAN–14 | Word2Vec Skip-gram, d=200, w=8 → W2V–DAN (h=256). |
| GLUE / SST-2 | W2V–DAN–15 | Word2Vec Skip-gram, d=200, w=10 → W2V–DAN (h=256). |
| GLUE / SST-2 | W2V–DAN–16 | Word2Vec Skip-gram, d=300, w=5 → W2V–DAN (h=256). |
| GLUE / SST-2 | W2V–DAN–17 | Word2Vec Skip-gram, d=300, w=8 → W2V–DAN (h=256). |
| GLUE / SST-2 | W2V–DAN–18 | Word2Vec Skip-gram, d=300, w=10 → W2V–DAN (h=256). |
| GLUE / SST-2 | RNN–BiGRU–1 | BERT tok., maxlen=66 → BiGRU (E=64, H=96, L=1). |
| GLUE / SST-2 | RNN–BiGRU–2 | BERT tok., maxlen=66 → BiGRU (E=64, H=96, L=2). |
| GLUE / SST-2 | RNN–BiGRU–3 | BERT tok., maxlen=66 → BiGRU (E=64, H=128, L=1). |
| GLUE / SST-2 | RNN–BiGRU–4 | BERT tok., maxlen=66 → BiGRU (E=64, H=128, L=2). |
| GLUE / SST-2 | RNN–BiGRU–5 | BERT tok., maxlen=66 → BiGRU (E=64, H=192, L=1). |
| GLUE / SST-2 | RNN–BiGRU–6 | BERT tok., maxlen=66 → BiGRU (E=64, H=192, L=2). |
| GLUE / SST-2 | RNN–BiGRU–7 | BERT tok., maxlen=66 → BiGRU (E=64, H=256, L=1). |

*(continued on next page)*

*(continued)*

| Dataset | Model | Description |
|---------|-------|-------------|
| GLUE / SST-2 | RNN–BiGRU–8 | BERT tok., maxlen=66 → BiGRU (E=64, H=256, L=2). |
| GLUE / SST-2 | RNN–BiGRU–9 | BERT tok., maxlen=66 → BiGRU (E=128, H=96, L=1). |
| GLUE / SST-2 | RNN–BiGRU–10 | BERT tok., maxlen=66 → BiGRU (E=128, H=96, L=2). |
| GLUE / SST-2 | RNN–BiGRU–11 | BERT tok., maxlen=66 → BiGRU (E=128, H=128, L=1). |
| GLUE / SST-2 | RNN–BiGRU–12 | BERT tok., maxlen=66 → BiGRU (E=128, H=128, L=2). |
| GLUE / SST-2 | RNN–BiGRU–13 | BERT tok., maxlen=66 → BiGRU (E=128, H=192, L=1). |
| GLUE / SST-2 | RNN–BiGRU–14 | BERT tok., maxlen=66 → BiGRU (E=128, H=192, L=2). |
| GLUE / SST-2 | RNN–BiGRU–15 | BERT tok., maxlen=66 → BiGRU (E=128, H=256, L=1). |
| GLUE / SST-2 | RNN–BiGRU–16 | BERT tok., maxlen=66 → BiGRU (E=128, H=256, L=2). |
| GLUE / SST-2 | RNN–BiGRU–17 | BERT tok., maxlen=66 → BiGRU (E=256, H=96, L=1). |
| GLUE / SST-2 | RNN–BiGRU–18 | BERT tok., maxlen=66 → BiGRU (E=256, H=96, L=2). |
| GLUE / SST-2 | RNN–BiGRU–19 | BERT tok., maxlen=66 → BiGRU (E=256, H=128, L=1). |
| GLUE / SST-2 | RNN–BiGRU–20 | BERT tok., maxlen=66 → BiGRU (E=256, H=128, L=2). |
| GLUE / SST-2 | RNN–BiGRU–21 | BERT tok., maxlen=66 → BiGRU (E=256, H=192, L=1). |
| GLUE / SST-2 | RNN–BiGRU–22 | BERT tok., maxlen=66 → BiGRU (E=256, H=192, L=2). |
| GLUE / SST-2 | RNN–BiGRU–23 | BERT tok., maxlen=66 → BiGRU (E=256, H=256, L=1). |
| GLUE / SST-2 | RNN–BiGRU–24 | BERT tok., maxlen=66 → BiGRU (E=256, H=256, L=2). |
| GLUE / SST-2 | TextCNN–1 | WordPiece (bert-base-uncased), L=66 → TextCNN (E=64, ks=(2,3,4), C=64). |
| GLUE / SST-2 | TextCNN–2 | WordPiece (bert-base-uncased), L=66 → TextCNN (E=64, ks=(3,4,5), C=64). |
| GLUE / SST-2 | TextCNN–3 | WordPiece (bert-base-uncased), L=66 → TextCNN (E=64, ks=(2,3,4,5), C=64). |
| GLUE / SST-2 | TextCNN–4 | WordPiece (bert-base-uncased), L=66 → TextCNN (E=128, ks=(2,3,4), C=64). |
| GLUE / SST-2 | TextCNN–5 | WordPiece (bert-base-uncased), L=66 → TextCNN (E=64, ks=(2,3,4), C=128). |
| GLUE / SST-2 | TextCNN–6 | WordPiece (bert-base-uncased), L=66 → TextCNN (E=128, ks=(3,4,5), C=64). |
| GLUE / SST-2 | TextCNN–7 | WordPiece (bert-base-uncased), L=66 → TextCNN (E=64, ks=(3,4,5), C=128). |
| GLUE / SST-2 | TextCNN–8 | WordPiece (bert-base-uncased), L=66 → TextCNN (E=128, ks=(2,3,4,5), C=64). |
| GLUE / SST-2 | TextCNN–9 | WordPiece (bert-base-uncased), L=66 → TextCNN (E=64, ks=(2,3,4,5), C=128). |
| GLUE / SST-2 | TextCNN–10 | WordPiece (bert-base-uncased), L=66 → TextCNN (E=256, ks=(2,3,4), C=64). |
| GLUE / SST-2 | TextCNN–11 | WordPiece (bert-base-uncased), L=66 → TextCNN (E=128, ks=(2,3,4), C=128). |
| GLUE / SST-2 | TextCNN–12 | WordPiece (bert-base-uncased), L=66 → TextCNN (E=64, ks=(2,3,4), C=256). |
| GLUE / SST-2 | TextCNN–13 | WordPiece (bert-base-uncased), L=66 → TextCNN (E=256, ks=(3,4,5), C=64). |

*(continued on next page)*

*(continued)*

| Dataset | Model | Description |
|---|---|---|
| GLUE / SST-2 | TextCNN–14 | WordPiece (bert-base-uncased), L=66 $\rightarrow$ TextCNN (E=128, ks=(3,4,5), C=128). |
| GLUE / SST-2 | TextCNN–15 | WordPiece (bert-base-uncased), L=66 $\rightarrow$ TextCNN (E=64, ks=(3,4,5), C=256). |
| GLUE / SST-2 | TextCNN–16 | WordPiece (bert-base-uncased), L=66 $\rightarrow$ TextCNN (E=256, ks=(2,3,4,5), C=64). |
| GLUE / SST-2 | TextCNN–17 | WordPiece (bert-base-uncased), L=66 $\rightarrow$ TextCNN (E=128, ks=(2,3,4,5), C=128). |
| GLUE / SST-2 | TextCNN–18 | WordPiece (bert-base-uncased), L=66 $\rightarrow$ TextCNN (E=64, ks=(2,3,4,5), C=256). |
| GLUE / SST-2 | TextCNN–19 | WordPiece (bert-base-uncased), L=66 $\rightarrow$ TextCNN (E=256, ks=(2,3,4), C=128). |
| GLUE / SST-2 | TextCNN–20 | WordPiece (bert-base-uncased), L=66 $\rightarrow$ TextCNN (E=128, ks=(2,3,4), C=256). |
| GLUE / SST-2 | TextCNN–21 | WordPiece (bert-base-uncased), L=66 $\rightarrow$ TextCNN (E=256, ks=(3,4,5), C=128). |
| GLUE / SST-2 | TextCNN–22 | WordPiece (bert-base-uncased), L=66 $\rightarrow$ TextCNN (E=128, ks=(3,4,5), C=256). |
| GLUE / SST-2 | TextCNN–23 | WordPiece (bert-base-uncased), L=66 $\rightarrow$ TextCNN (E=256, ks=(2,3,4,5), C=128). |
| GLUE / SST-2 | TextCNN–24 | WordPiece (bert-base-uncased), L=66 $\rightarrow$ TextCNN (E=128, ks=(2,3,4,5), C=256). |
| GLUE / SST-2 | TextCNN–25 | WordPiece (bert-base-uncased), L=66 $\rightarrow$ TextCNN (E=256, ks=(2,3,4), C=256). |
| GLUE / SST-2 | TextCNN–26 | WordPiece (bert-base-uncased), L=66 $\rightarrow$ TextCNN (E=256, ks=(3,4,5), C=256). |
| GLUE / SST-2 | TextCNN–27 | WordPiece (bert-base-uncased), L=66 $\rightarrow$ TextCNN (E=256, ks=(2,3,4,5), C=256). |
| GLUE / SST-2 | SBERT–GCN–1 | SBERT (all-mpnet-base-v2, d=768); doc–doc top-$k$=5 $\rightarrow$ GCN-1L ($h_1$=64). |
| GLUE / SST-2 | SBERT–GCN–2 | SBERT (all-mpnet-base-v2, d=768); doc–doc top-$k$=5 $\rightarrow$ GCN-1L ($h_1$=128). |
| GLUE / SST-2 | SBERT–GCN–3 | SBERT (all-mpnet-base-v2, d=768); doc–doc top-$k$=5 $\rightarrow$ GCN-1L ($h_1$=256). |
| GLUE / SST-2 | SBERT–GCN–4 | SBERT (all-mpnet-base-v2, d=768); doc–doc top-$k$=5 $\rightarrow$ GCN-2L ($h_1$=256, $h_2$=256). |
| GLUE / SST-2 | SBERT–GCN–5 | SBERT (all-mpnet-base-v2, d=768); doc–doc top-$k$=5 $\rightarrow$ GCN-2L ($h_1$=256, $h_2$=128). |
| GLUE / SST-2 | SBERT–GCN–6 | SBERT (all-mpnet-base-v2, d=768); doc–doc top-$k$=5 $\rightarrow$ GCN-2L ($h_1$=256, $h_2$=64). |
| GLUE / SST-2 | SBERT–GCN–7 | SBERT (all-mpnet-base-v2, d=768); doc–doc top-$k$=5 $\rightarrow$ GCN-2L ($h_1$=128, $h_2$=128). |
| GLUE / SST-2 | SBERT–GCN–8 | SBERT (all-mpnet-base-v2, d=768); doc–doc top-$k$=5 $\rightarrow$ GCN-2L ($h_1$=128, $h_2$=64). |
| GLUE / SST-2 | SBERT–GCN–9 | SBERT (all-mpnet-base-v2, d=768); doc–doc top-$k$=5 $\rightarrow$ GCN-2L ($h_1$=64, $h_2$=64). |

*(continued on next page)*

| Dataset | Model | Description |
|---|---|---|
| GLUE / SST-2 | SBERT–GCN–10 | SBERT (all-mpnet-base-v2, d=768); doc–doc top-$k$=10 $\rightarrow$ GCN-1L ($h_1$=64). |
| GLUE / SST-2 | SBERT–GCN–11 | SBERT (all-mpnet-base-v2, d=768); doc–doc top-$k$=10 $\rightarrow$ GCN-1L ($h_1$=128). |
| GLUE / SST-2 | SBERT–GCN–12 | SBERT (all-mpnet-base-v2, d=768); doc–doc top-$k$=10 $\rightarrow$ GCN-1L ($h_1$=256). |
| GLUE / SST-2 | SBERT–GCN–13 | SBERT (all-mpnet-base-v2, d=768); doc–doc top-$k$=10 $\rightarrow$ GCN-2L ($h_1$=256, $h_2$=256). |
| GLUE / SST-2 | SBERT–GCN–14 | SBERT (all-mpnet-base-v2, d=768); doc–doc top-$k$=10 $\rightarrow$ GCN-2L ($h_1$=256, $h_2$=128). |
| GLUE / SST-2 | SBERT–GCN–15 | SBERT (all-mpnet-base-v2, d=768); doc–doc top-$k$=10 $\rightarrow$ GCN-2L ($h_1$=256, $h_2$=64). |
| GLUE / SST-2 | SBERT–GCN–16 | SBERT (all-mpnet-base-v2, d=768); doc–doc top-$k$=10 $\rightarrow$ GCN-2L ($h_1$=128, $h_2$=128). |
| GLUE / SST-2 | SBERT–GCN–17 | SBERT (all-mpnet-base-v2, d=768); doc–doc top-$k$=10 $\rightarrow$ GCN-2L ($h_1$=128, $h_2$=64). |
| GLUE / SST-2 | SBERT–GCN–18 | SBERT (all-mpnet-base-v2, d=768); doc–doc top-$k$=10 $\rightarrow$ GCN-2L ($h_1$=64, $h_2$=64). |
| GLUE / SST-2 | SBERT–GCN–19 | SBERT (all-mpnet-base-v2, d=768); doc–doc top-$k$=20 $\rightarrow$ GCN-1L ($h_1$=64). |
| GLUE / SST-2 | SBERT–GCN–20 | SBERT (all-mpnet-base-v2, d=768); doc–doc top-$k$=20 $\rightarrow$ GCN-1L ($h_1$=128). |
| GLUE / SST-2 | SBERT–GCN–21 | SBERT (all-mpnet-base-v2, d=768); doc–doc top-$k$=20 $\rightarrow$ GCN-1L ($h_1$=256). |
| GLUE / SST-2 | SBERT–GCN–22 | SBERT (all-mpnet-base-v2, d=768); doc–doc top-$k$=20 $\rightarrow$ GCN-2L ($h_1$=256, $h_2$=256). |
| GLUE / SST-2 | SBERT–GCN–23 | SBERT (all-mpnet-base-v2, d=768); doc–doc top-$k$=20 $\rightarrow$ GCN-2L ($h_1$=256, $h_2$=128). |
| GLUE / SST-2 | SBERT–GCN–24 | SBERT (all-mpnet-base-v2, d=768); doc–doc top-$k$=20 $\rightarrow$ GCN-2L ($h_1$=256, $h_2$=64). |
| GLUE / SST-2 | SBERT–GCN–25 | SBERT (all-mpnet-base-v2, d=768); doc–doc top-$k$=20 $\rightarrow$ GCN-2L ($h_1$=128, $h_2$=128). |
| GLUE / SST-2 | SBERT–GCN–26 | SBERT (all-mpnet-base-v2, d=768); doc–doc top-$k$=20 $\rightarrow$ GCN-2L ($h_1$=128, $h_2$=64). |
| GLUE / SST-2 | SBERT–GCN–27 | SBERT (all-mpnet-base-v2, d=768); doc–doc top-$k$=20 $\rightarrow$ GCN-2L ($h_1$=64, $h_2$=64). |
| GLUE / SST-2 | BERT–1 | BERT (Devlin et al., 2019) (`bert-base-uncased`). |
| GLUE / SST-2 | BERT–2 | DISTILBERT (Sanh et al., 2019) (`distilbert-base-uncased`). |
| GLUE / SST-2 | BERT–3 | ALBERT (Lan et al., 2020) (`albert-base-v2`). |
| GLUE / SST-2 | BERT–4 | MINILM (Wang et al., 2020) (`microsoft/MiniLM-L6-H384`). |
| GLUE / SST-2 | BERT–5 | CONVBERT (Jiang et al., 2020) (`convbert-base`). |
| GLUE / SST-2 | BERT–6 | DEBERTA V3 (He et al., 2021) (`microsoft/deberta-v3-base`). |
| GLUE / SST-2 | BERT–7 | ROBERTA (Liu et al., 2019) (`roberta-base`). |

| Dataset | Model | Description |
|---|---|---|
| GLUE / SST-2 | BERT–8 | XLNET (Yang et al., 2019) (`xlnet-base-cased`). |

Table 6: Models used on the CIFAR-10 dataset.

| Dataset | Model | Description |
|---|---|---|
| CIFAR-10 | HOG–SVM-1 | HOG (gray; pixels/cell $4\times4$, cells/block $2\times2$, orientations 9); features L2-Hys; classifier: LinearSVC (OvR, $C{=}1$, L2). |
| CIFAR-10 | HOG–SVM-2 | HOG (gray; $8\times8$ ppc, $2\times2$ cpb, 9 ori); L2-Hys; classifier: LinearSVC (OvR, $C{=}1$, L2). |
| CIFAR-10 | HOG–SVM-3 | HOG (RGB, per-channel concat; $4\times4$ ppc, $2\times2$ cpb, 9 ori); L2-Hys; classifier: LinearSVC (OvR, $C{=}1$, L2). |
| CIFAR-10 | HOG–SVM-4 | HOG (RGB, per-channel concat; $8\times8$ ppc, $2\times2$ cpb, 9 ori); L2-Hys; classifier: LinearSVC (OvR, $C{=}1$, L2). |
| CIFAR-10 | SimpleCNN-1 | 3 stages: (Conv–BN–ReLU $3\times3$)$\times2$ + MP(2) per stage; base$=16$; head: GAP $\rightarrow$ MLP($h{=}64$); dropout$=0.1$. |
| CIFAR-10 | SimpleCNN-2 | 3 stages: (Conv–BN–ReLU $3\times3$)$\times2$ + MP(2) per stage; base$=32$; head: GAP $\rightarrow$ MLP($h{=}128$); dropout$=0.1$. |
| CIFAR-10 | SimpleCNN-3 | 3 stages: (Conv–BN–ReLU $3\times3$)$\times2$ + MP(2) per stage; base$=48$; head: GAP $\rightarrow$ MLP($h{=}256$); dropout$=0.2$. |
| CIFAR-10 | SimpleCNN-4 | 3 stages: (Conv–BN–ReLU $3\times3$)$\times2$ + MP(2) per stage; base$=64$; head: GAP $\rightarrow$ MLP($h{=}256$); dropout$=0.3$. |
| CIFAR-10 | SimpleCNN-5 | 3 stages: (Conv–BN–ReLU $3\times3$)$\times2$ + MP(2) per stage; base$=96$; head: GAP $\rightarrow$ MLP($h{=}512$); dropout$=0.4$. |
| CIFAR-10 | SimpleCNN-6 | 3 stages: (Conv–BN–ReLU $3\times3$)$\times2$ + MP(2) per stage; base$=128$; head: GAP $\rightarrow$ MLP($h{=}512$); dropout$=0.5$. |
| CIFAR-10 | SimpleCNN-7 | 3 stages: (Conv–BN–ReLU $3\times3$)$\times2$ + MP(2) per stage; base$=64$; head: GAP $\rightarrow$ Linear(10); no hidden MLP; dropout$=0.0$. |
| CIFAR-10 | SimpleCNN-8 | 3 stages: (Conv–BN–ReLU $3\times3$)$\times2$ + MP(2) per stage; base$=64$; head: GAP $\rightarrow$ MLP($h{=}1024$); dropout$=0.3$. |
| CIFAR-10 | MLP-1 | Flatten $\rightarrow$ MLP($h = []$; i.e., no hidden). |
| CIFAR-10 | MLP-2 | Flatten $\rightarrow$ MLP($h = [128]$). |
| CIFAR-10 | MLP-3 | Flatten $\rightarrow$ MLP($h = [256]$). |
| CIFAR-10 | MLP-4 | Flatten $\rightarrow$ MLP($h = [256, 128]$). |
| CIFAR-10 | MLP-5 | Flatten $\rightarrow$ MLP($h = [512, 256]$). |
| CIFAR-10 | MLP-6 | Flatten $\rightarrow$ MLP($h = [1024, 512]$). |
| CIFAR-10 | MLP-7 | Flatten $\rightarrow$ MLP($h = [1024, 512, 256]$). |
| CIFAR-10 | MLP-8 | Flatten $\rightarrow$ MLP($h = [1024, 512, 256, 128]$). |
| CIFAR-10 | ResNet-1 | Stem: Conv $7\times7$, $s{=}2 \rightarrow$ MaxPool $3\times3$, $s{=}2$; 4 stages: BasicBlock $\{2, 2, 2, 2\}$, widths $\{64, 128, 256, 512\}$; head: GAP $\rightarrow$ Linear(10). He et al. (2016) |
| CIFAR-10 | ResNet-2 | Stem: Conv $7\times7$, $s{=}2 \rightarrow$ MaxPool $3\times3$, $s{=}2$; 4 stages: BasicBlock $\{3, 4, 6, 3\}$, widths $\{64, 128, 256, 512\}$; head: GAP $\rightarrow$ Linear(10). |

*(continued)*

| Dataset | Model | Description |
|---|---|---|
| CIFAR-10 | ResNet-3 | Stem: Conv 7×7, $s=2$ → MaxPool 3×3, $s=2$; 4 stages: Bottleneck $\{3, 4, 6, 3\}$, planes $\{64, 128, 256, 512\}$ (out $\{256, 512, 1024, 2048\}$); head: GAP → Linear(10). |
| CIFAR-10 | ResNet-4 | Stem: Conv 7×7, $s=2$ → MaxPool 3×3, $s=2$; 4 stages: Bottleneck $\{3, 8, 36, 3\}$, planes $\{64, 128, 256, 512\}$ (out $\{256, 512, 1024, 2048\}$); head: GAP → Linear(10). |
| CIFAR-10 | ResNet-5 | Stem: Conv 7×7, $s=2$ → MaxPool 3×3, $s=2$; 4 stages: Bottleneck $\{3, 4, 6, 3\}$, cardinality 32, base width 4; head: GAP → Linear(10). |
| CIFAR-10 | ResNet-6 | Stem: Conv 7×7, $s=2$ → MaxPool 3×3, $s=2$; 4 stages: Bottleneck $\{3, 4, 23, 3\}$, cardinality 32, base width 8; head: GAP → Linear(10). |
| CIFAR-10 | ResNet-7 | Stem: Conv 7×7, $s=2$ → MaxPool 3×3, $s=2$; 4 stages: Bottleneck $\{3, 4, 23, 3\}$, widths doubled (planes $\{128, 256, 512, 1024\}$); head: GAP → Linear(10). |
| CIFAR-10 | MobileNetV2-1 | Stack of InvertedResidual (DW 3×3, expansion $t=6$); width multiplier $\alpha=0.35$; head: 1×1 conv → GAP → Linear(10). Sandler et al. (2018) |
| CIFAR-10 | MobileNetV2-2 | Stack of InvertedResidual (DW 3×3, $t=6$); $\alpha=0.50$; head: 1×1 conv → GAP → Linear(10). |
| CIFAR-10 | MobileNetV2-3 | Stack of InvertedResidual (DW 3×3, $t=6$); $\alpha=0.75$; head: 1×1 conv → GAP → Linear(10). |
| CIFAR-10 | MobileNetV2-4 | Stack of InvertedResidual (DW 3×3, $t=6$); $\alpha=1.0$; head: 1×1 conv → GAP → Linear(10). |
| CIFAR-10 | MobileNetV2-5 | Stack of InvertedResidual (DW 3×3, $t=6$); $\alpha=1.1$d; head: 1×1 conv → GAP → Linear(10). |
| CIFAR-10 | MobileNetV2-6 | Stack of InvertedResidual (DW 3×3, $t=6$); $\alpha=1.4$; head: 1×1 conv → GAP → Linear(10). |
| CIFAR-10 | ShuffleNetV2-1 | Shuffle units; $s=0.5$; head: 1×1 conv → GAP → Linear(10). Ma et al. (2018) |
| CIFAR-10 | ShuffleNetV2-2 | Shuffle units; $s=1.0$; head: 1×1 conv → GAP → Linear(10). |
| CIFAR-10 | ShuffleNetV2-3 | Shuffle units; $s=1.5$; head: 1×1 conv → GAP → Linear(10). |
| CIFAR-10 | ShuffleNetV2-4 | Shuffle units; $s=2.0$; head: 1×1 conv → GAP → Linear(10). |
| CIFAR-10 | GhostNet-1 | Ghost bottlenecks (cheap ops generate ghost features) with SE; width $\alpha=0.5$; head: 1×1 conv → GAP → Linear(10). Han et al. (2020) |
| CIFAR-10 | GhostNet-2 | Ghost bottlenecks + SE; $\alpha=1.0$; head: 1×1 conv → GAP → Linear(10). |
| CIFAR-10 | GhostNet-3 | Ghost bottlenecks + SE; $\alpha=1.3$; head: 1×1 conv → GAP → Linear(10). |
| CIFAR-10 | GhostNet-4 | GhostNetV2 (decoupled modules, improved feature generation); $\alpha=1.0$; head: 1×1 conv → GAP → Linear(10). |
| CIFAR-10 | GhostNet-5 | GhostNetV2; $\alpha=1.3$; head: 1×1 conv → GAP → Linear(10). |
| CIFAR-10 | GhostNet-6 | GhostNetV2; $\alpha=1.6$; head: 1×1 conv → GAP → Linear(10). |
| CIFAR-10 | DeiT-1 | PatchEmbed 16×16 ($s=16$) → Transformer encoder ($L=12$, $d=192$, heads= 3); head: LN → Linear(10). Touvron et al. (2021) |

*(continued on next page)*

*(continued)*

| Dataset | Model | Description |
|---|---|---|
| CIFAR-10 | DeiT-2 | PatchEmbed $16\times16$ ($s{=}16$) $\rightarrow$ Transformer encoder ($L{=}12$, $d{=}384$, heads$=6$); head: LN $\rightarrow$ Linear(10). |
| CIFAR-10 | DeiT-3 | PatchEmbed $16\times16$ ($s{=}16$) $\rightarrow$ Transformer encoder ($L{=}12$, $d{=}768$, heads$=12$); head: LN $\rightarrow$ Linear(10). |
| CIFAR-10 | Swin-1 | PatchEmbed $4\times4$ ($s{=}4$) $\rightarrow$ 4 stages (window MSA, win$=7$); depths $\{2,2,6,2\}$, dims $\{96,192,384,768\}$; head: GAP $\rightarrow$ Linear(10). Liu et al. (2021) |
| CIFAR-10 | Swin-2 | PatchEmbed $4\times4$ ($s{=}4$) $\rightarrow$ 4 stages (window MSA, win$=7$); depths $\{2,2,18,2\}$, dims $\{96,192,384,768\}$; head: GAP $\rightarrow$ Linear(10). |
| CIFAR-10 | ConvNeXt-1 | Stem: Conv $4\times4$ ($s{=}4$) patchify $\rightarrow$ stages with blocks $\{3,3,9,3\}$, dims $\{96,192,384,768\}$; head: GAP $\rightarrow$ Linear(10). Liu et al. (2022) |
| CIFAR-10 | ConvNeXt-2 | Stem: Conv $4\times4$ ($s{=}4$) patchify $\rightarrow$ stages with blocks $\{3,3,27,3\}$, dims $\{96,192,384,768\}$; head: GAP $\rightarrow$ Linear(10). |
| CIFAR-10 | ConvNeXt-3 | Stem: Conv $4\times4$ ($s{=}4$) patchify $\rightarrow$ stages with blocks $\{3,3,27,3\}$, dims $\{128,256,512,1024\}$; head: GAP $\rightarrow$ Linear(10). |

Table 7: Models used on the UrbanSound8K dataset.

| Dataset | Model | Description |
|---|---|---|
| UrbanSound8K | TinyCNN–1 | MelSpec (SR=8k, $D{=}3.0$s, $N_{\mathrm{mel}}{=}40$, win$=25$ms, hop$=10$ms) $\rightarrow$ dB; Backbone: $\{$Conv–BN–ReLU$\}\times2$ @$\{24\}$ $\rightarrow$ MP(2) $\rightarrow$ $\{$Conv–BN–ReLU$\}\times2$ @$\{48\}$ $\rightarrow$ MP(2) $\rightarrow$ $\{$Conv–BN–ReLU$\}\times2$ @$\{96\}$ $\rightarrow$ MP(2); Head: GAP $\rightarrow$ Linear(10). |
| UrbanSound8K | TinyCNN–2 | MelSpec (SR=16k, $D{=}4.0$s, $N_{\mathrm{mel}}{=}64$, win$=25$ms, hop$=10$ms) $\rightarrow$ dB; Backbone: $\{$Conv–BN–ReLU$\}\times2$ @$\{32\}$ $\rightarrow$ MP(2) $\rightarrow$ $\{$Conv–BN–ReLU$\}\times2$ @$\{64\}$ $\rightarrow$ MP(2) $\rightarrow$ $\{$Conv–BN–ReLU$\}\times2$ @$\{128\}$ $\rightarrow$ MP(2); Head: GAP $\rightarrow$ Linear(10). |
| UrbanSound8K | TinyCNN–3 | MelSpec (SR=16k, $D{=}3.5$s, $N_{\mathrm{mel}}{=}64$, win$=25$ms, hop$=10$ms) $\rightarrow$ dB; Backbone: $\{$Conv–BN–ReLU$\}\times2$ @$\{32\}$ $\rightarrow$ MP(2) $\rightarrow$ $\{$Conv–BN–ReLU$\}\times2$ @$\{64\}$ $\rightarrow$ MP(2) $\rightarrow$ $\{$Conv–BN–ReLU$\}\times2$ @$\{128\}$ $\rightarrow$ MP(2); Head: GAP $\rightarrow$ Linear(10). |
| UrbanSound8K | TinyCNN–4 | MelSpec (SR=22.05k, $D{=}4.0$s, $N_{\mathrm{mel}}{=}96$, win$=32$ms, hop$=10$ms) $\rightarrow$ dB; Backbone: $\{$Conv–BN–ReLU$\}\times2$ @$\{48\}$ $\rightarrow$ MP(2) $\rightarrow$ $\{$Conv–BN–ReLU$\}\times2$ @$\{96\}$ $\rightarrow$ MP(2) $\rightarrow$ $\{$Conv–BN–ReLU$\}\times2$ @$\{192\}$ $\rightarrow$ MP(2); Head: GAP $\rightarrow$ Linear(10). |
| UrbanSound8K | TinyCNN–5 | MelSpec (SR=8k, $D{=}4.0$s, $N_{\mathrm{mel}}{=}40$, win$=25$ms, hop$=10$ms) $\rightarrow$ dB; Backbone: $\{$Conv–BN–ReLU$\}\times2$ @$\{16\}$ $\rightarrow$ MP(2) $\rightarrow$ $\{$Conv–BN–ReLU$\}\times2$ @$\{32\}$ $\rightarrow$ MP(2) $\rightarrow$ $\{$Conv–BN–ReLU$\}\times2$ @$\{64\}$ $\rightarrow$ MP(2); Head: GAP $\rightarrow$ Linear(10). |

*(continued on next page)*

*(continued)*

| Dataset | Model | Description |
|---------|-------|-------------|
| UrbanSound8K | TinyCNN–6 | MelSpec (SR=16k, $D$=4.0s, $N_{\mathrm{mel}}$=64, win= 25ms, hop= 10ms) → dB; Backbone: {Conv–BN–ReLU}×2 @{48} → MP(2) → {Conv–BN–ReLU}×2 @{96} → MP(2) → {Conv–BN–ReLU}×2 @{192} → MP(2); Head: GAP → Linear(10). |
| UrbanSound8K | TinyCNN–7 | MelSpec (SR=16k, $D$=4.0s, $N_{\mathrm{mel}}$=64, win= 25ms, hop= 5ms) → dB; Backbone: {Conv–BN–ReLU}×2 @{32} → MP(2) → {Conv–BN–ReLU}×2 @{64} → MP(2) → {Conv–BN–ReLU}×2 @{128} → MP(2); Head: GAP → Linear(10). |
| UrbanSound8K | TinyCNN–8 | MelSpec (SR=16k, $D$=4.0s, $N_{\mathrm{mel}}$=64, win= 25ms, hop= 10ms) → dB; Backbone: {Conv–BN–ReLU}×2 @{32} → MP(2) → {Conv–BN–ReLU}×2 @{64} → MP(2) → {Conv–BN–ReLU}×2 @{128} → MP(2); Head: GAP → Linear(10). |
| UrbanSound8K | CRNN–1 | MelSpec (SR=8k, $D$=3.0s, $N_{\mathrm{mel}}$=40, win= 25ms, hop= 10ms) → dB; Backbone: {Conv–BN–ReLU}×2@{16} → MP(2) → {Conv–BN–ReLU}×2@{32} → MP(2) → {Conv–BN–ReLU}×2@{64} → MP(2); Recurrent: BiGRU($H$=64, $L$=1); Head: last-step → Linear(10). |
| UrbanSound8K | CRNN–2 | MelSpec (SR=16k, $D$=4.0s, $N_{\mathrm{mel}}$=64, win= 25ms, hop= 10ms) → dB; Backbone: {Conv–BN–ReLU}×2@{32} → MP(2) → {Conv–BN–ReLU}×2@{64} → MP(2) → {Conv–BN–ReLU}×2@{128} → MP(2); Recurrent: BiGRU($H$=128, $L$=1); Head: last-step → Linear(10). |
| UrbanSound8K | CRNN–3 | MelSpec (SR=16k, $D$=3.5s, $N_{\mathrm{mel}}$=64, win= 25ms, hop= 10ms) → dB; Backbone: {Conv–BN–ReLU}×2@{32} → MP(2) → {Conv–BN–ReLU}×2@{64} → MP(2) → {Conv–BN–ReLU}×2@{128} → MP(2); Recurrent: BiGRU($H$=128, $L$=1); Head: last-step → Linear(10). |
| UrbanSound8K | CRNN–4 | MelSpec (SR=22.05k, $D$=4.0s, $N_{\mathrm{mel}}$=96, win= 32ms, hop= 10ms) → dB; Backbone: {Conv–BN–ReLU}×2@{48} → MP(2) → {Conv–BN–ReLU}×2@{96} → MP(2) → {Conv–BN–ReLU}×2@{192} → MP(2); Recurrent: BiGRU($H$=256, $L$=2); Head: last-step → Linear(10). |
| UrbanSound8K | CRNN–5 | MelSpec (SR=8k, $D$=4.0s, $N_{\mathrm{mel}}$=40, win= 25ms, hop= 10ms) → dB; Backbone: {Conv–BN–ReLU}×2@{16} → MP(2) → {Conv–BN–ReLU}×2@{32} → MP(2) → {Conv–BN–ReLU}×2@{64} → MP(2); Recurrent: BiGRU($H$=64, $L$=1); Head: last-step → Linear(10). |
| UrbanSound8K | CRNN–6 | MelSpec (SR=16k, $D$=4.0s, $N_{\mathrm{mel}}$=64, win= 25ms, hop= 10ms) → dB; Backbone: {Conv–BN–ReLU}×2@{48} → MP(2) → {Conv–BN–ReLU}×2@{96} → MP(2) → {Conv–BN–ReLU}×2@{192} → MP(2); Recurrent: BiGRU($H$=256, $L$=1); Head: last-step → Linear(10). |

*(continued on next page)*

| | | (continued) |
|---|---|---|
| **Dataset** | **Model** | **Description** |
| UrbanSound8K | CRNN–7 | MelSpec (SR=16k, $D$=4.0s, $N_{\mathrm{mel}}$=64, win= 25ms, hop= 5ms) $\rightarrow$ dB; Backbone: {Conv–BN–ReLU}$\times$2@{32} $\rightarrow$ MP(2) $\rightarrow$ {Conv–BN–ReLU}$\times$2@{64} $\rightarrow$ MP(2) $\rightarrow$ {Conv–BN–ReLU}$\times$2@{128} $\rightarrow$ MP(2); Recurrent: BiGRU($H$=128, $L$=1); Head: last-step $\rightarrow$ Linear(10). |
| UrbanSound8K | CRNN–8 | MelSpec (SR=16k, $D$=4.0s, $N_{\mathrm{mel}}$=64, win= 25ms, hop= 10ms) $\rightarrow$ dB; Backbone: {Conv–BN–ReLU}$\times$2@{32} $\rightarrow$ MP(2) $\rightarrow$ {Conv–BN–ReLU}$\times$2@{64} $\rightarrow$ MP(2) $\rightarrow$ {Conv–BN–ReLU}$\times$2@{128} $\rightarrow$ MP(2); Recurrent: BiGRU($H$=128, $L$=2); Head: last-step $\rightarrow$ Linear(10). |
| UrbanSound8K | CRNN–9 | MelSpec (SR=16k, $D$=4.0s, $N_{\mathrm{mel}}$=128, win= 32ms, hop= 10ms) $\rightarrow$ dB; Backbone: {Conv–BN–ReLU}$\times$2@{64} $\rightarrow$ MP(2) $\rightarrow$ {Conv–BN–ReLU}$\times$2@{128} $\rightarrow$ MP(2) $\rightarrow$ {Conv–BN–ReLU}$\times$2@{256} $\rightarrow$ MP(2); Recurrent: BiGRU($H$=512, $L$=2); Head: last-step $\rightarrow$ Linear(10). |
| UrbanSound8K | ResNet–1 | MelSpec (SR=16k, $D$=4.0s, $N_{\mathrm{mel}}$=64, win= 25ms, hop= 10ms) $\rightarrow$ dB; Backbone: ResNet-18 (BasicBlock), stages $[2, 2, 2, 2]$; width_mult= 0.75; conv1 stride= 1; maxpool: off; Head: GAP $\rightarrow$ Linear(10). |
| UrbanSound8K | ResNet–2 | MelSpec (SR=16k, $D$=4.0s, $N_{\mathrm{mel}}$=64, win= 25ms, hop= 10ms) $\rightarrow$ dB; Backbone: ResNet-18 (BasicBlock), stages $[2, 2, 2, 2]$; width_mult= 1.00; conv1 stride= 2; maxpool: on; Head: GAP $\rightarrow$ Linear(10). |
| UrbanSound8K | ResNet–3 | MelSpec (SR=16k, $D$=4.0s, $N_{\mathrm{mel}}$=64, win= 25ms, hop= 10ms) $\rightarrow$ dB; Backbone: ResNet-18 (BasicBlock), stages $[2, 2, 2, 2]$; width_mult= 1.25; conv1 stride= 1; maxpool: on; Head: GAP $\rightarrow$ Linear(10). |
| UrbanSound8K | ResNet–4 | MelSpec (SR=16k, $D$=4.0s, $N_{\mathrm{mel}}$=64, win= 25ms, hop= 10ms) $\rightarrow$ dB; Backbone: ResNet-34 (BasicBlock), stages $[3, 4, 6, 3]$; width_mult= 0.75; conv1 stride= 1; maxpool: off; Head: GAP $\rightarrow$ Linear(10). |
| UrbanSound8K | ResNet–5 | MelSpec (SR=16k, $D$=4.0s, $N_{\mathrm{mel}}$=64, win= 25ms, hop= 10ms) $\rightarrow$ dB; Backbone: ResNet-34 (BasicBlock), stages $[3, 4, 6, 3]$; width_mult= 1.00; conv1 stride= 2; maxpool: on; Head: GAP $\rightarrow$ Linear(10). |
| UrbanSound8K | ResNet–6 | MelSpec (SR=16k, $D$=4.0s, $N_{\mathrm{mel}}$=64, win= 25ms, hop= 10ms) $\rightarrow$ dB; Backbone: ResNet-34 (BasicBlock), stages $[3, 4, 6, 3]$; width_mult= 1.25; conv1 stride= 1; maxpool: on; Head: GAP $\rightarrow$ Linear(10). |
| UrbanSound8K | ResNet–7 | MelSpec (SR=16k, $D$=4.0s, $N_{\mathrm{mel}}$=64, win= 25ms, hop= 10ms) $\rightarrow$ dB; Backbone: ResNet-50 (Bottleneck), stages $[3, 4, 6, 3]$; width_mult= 0.75; conv1 stride= 1; maxpool: off; Head: GAP $\rightarrow$ Linear(10). |
| UrbanSound8K | ResNet–8 | MelSpec (SR=16k, $D$=4.0s, $N_{\mathrm{mel}}$=64, win= 25ms, hop= 10ms) $\rightarrow$ dB; Backbone: ResNet-50 (Bottleneck), stages $[3, 4, 6, 3]$; width_mult= 1.00; conv1 stride= 2; maxpool: on; Head: GAP $\rightarrow$ Linear(10). |

*(continued)*

| Dataset | Model | Description |
|---|---|---|
| UrbanSound8K | ResNet–9 | MelSpec (SR=16k, $D$=4.0s, $N_{\mathrm{mel}}$=64, win= 25ms, hop= 10ms) $\to$ dB; Backbone: ResNet-50 (Bottleneck), stages $[3, 4, 6, 3]$; width_mult= 1.25; conv1 stride= 1; maxpool: on; Head: GAP $\to$ Linear(10). |
| UrbanSound8K | PANN–1 | CNN6: 3 stages, (Conv–BN–ReLU $3{\times}3$)$\times 2$ + MP($2{\times}2$) per stage; width_mult= 0.75; frequency average $\to$ temporal gated-attention $\to$ FC. Kong et al. (2020) |
| UrbanSound8K | PANN–2 | CNN6: 3 stages, (Conv–BN–ReLU $3{\times}3$)$\times 2$ + MP($2{\times}2$) per stage; width_mult= 1.00; frequency average $\to$ temporal gated-attention $\to$ FC. |
| UrbanSound8K | PANN–3 | CNN6: 3 stages, (Conv–BN–ReLU $3{\times}3$)$\times 2$ + MP($2{\times}2$) per stage; width_mult= 1.25; frequency average $\to$ temporal gated-attention $\to$ FC. |
| UrbanSound8K | PANN–4 | CNN10: 5 stages, (Conv–BN–ReLU $3{\times}3$)$\times 2$ + MP($2{\times}2$) per stage; width_mult= 0.75; frequency average $\to$ temporal gated-attention $\to$ FC. |
| UrbanSound8K | PANN–5 | CNN10: 5 stages, (Conv–BN–ReLU $3{\times}3$)$\times 2$ + MP($2{\times}2$) per stage; width_mult= 1.00; frequency average $\to$ temporal gated-attention $\to$ FC. |
| UrbanSound8K | PANN–6 | CNN10: 5 stages, (Conv–BN–ReLU $3{\times}3$)$\times 2$ + MP($2{\times}2$) per stage; width_mult= 1.25; frequency average $\to$ temporal gated-attention $\to$ FC. |
| UrbanSound8K | AST–1 | PatchEmbed ($p_f$=16, $p_t$=16); Transformer $d$=192, depth= 6, heads= 3; 2D pos-enc + [CLS]; LN $\to$ Linear. Gong et al. (2021) |
| UrbanSound8K | AST–2 | PatchEmbed ($p_f$=16, $p_t$=8); Transformer $d$=192, depth= 6, heads= 3; 2D pos-enc + [CLS]; LN $\to$ Linear. |
| UrbanSound8K | AST–3 | PatchEmbed ($p_f$=32, $p_t$=16); Transformer $d$=192, depth= 6, heads= 3; 2D pos-enc + [CLS]; LN $\to$ Linear. |
| UrbanSound8K | AST–4 | PatchEmbed ($p_f$=24, $p_t$=24); Transformer $d$=192, depth= 6, heads= 3; 2D pos-enc + [CLS]; LN $\to$ Linear. |
| UrbanSound8K | AST–5 | PatchEmbed ($p_f$=16, $p_t$=16); Transformer $d$=384, depth= 8, heads= 6; 2D pos-enc + [CLS]; LN $\to$ Linear. |
| UrbanSound8K | AST–6 | PatchEmbed ($p_f$=16, $p_t$=8); Transformer $d$=384, depth= 8, heads= 6; 2D pos-enc + [CLS]; LN $\to$ Linear. |
| UrbanSound8K | AST–7 | PatchEmbed ($p_f$=32, $p_t$=16); Transformer $d$=384, depth= 8, heads= 6; 2D pos-enc + [CLS]; LN $\to$ Linear. |
| UrbanSound8K | AST–8 | PatchEmbed ($p_f$=24, $p_t$=24); Transformer $d$=384, depth= 8, heads= 6; 2D pos-enc + [CLS]; LN $\to$ Linear. |
| UrbanSound8K | LSTM–1 | Log-Mel $\to$ framewise LN $\to$ Linear($F \to 128$) $\to$ BiLSTM($H$=128, $L$=1) $\to$ Last-pool $\to$ LN $\to$ FC. |
| UrbanSound8K | LSTM–2 | Log-Mel $\to$ framewise LN $\to$ Linear($F \to 128$) $\to$ BiLSTM($H$=128, $L$=2) $\to$ Attn-pool $\to$ LN $\to$ FC. |
| UrbanSound8K | LSTM–3 | Log-Mel $\to$ framewise LN $\to$ Linear($F \to 192$) $\to$ BiLSTM($H$=128, $L$=2) $\to$ Mean-pool $\to$ LN $\to$ FC. |
| UrbanSound8K | LSTM–4 | Log-Mel $\to$ framewise LN $\to$ Linear($F \to 192$) $\to$ BiLSTM($H$=256, $L$=2) $\to$ Attn-pool $\to$ LN $\to$ FC. |

*(continued on next page)*

*(continued)*

| Dataset | Model | Description |
|---|---|---|
| UrbanSound8K | LSTM–5 | Log-Mel $\rightarrow$ framewise LN $\rightarrow$ Linear($F \rightarrow 256$) $\rightarrow$ BiLSTM($H\!=\!256$, $L\!=\!2$) $\rightarrow$ Last-pool $\rightarrow$ LN $\rightarrow$ FC. |
| UrbanSound8K | LSTM–6 | Log-Mel $\rightarrow$ framewise LN $\rightarrow$ Linear($F \rightarrow 192$) $\rightarrow$ BiLSTM($H\!=\!256$, $L\!=\!3$) $\rightarrow$ Attn-pool $\rightarrow$ LN $\rightarrow$ FC. |
| UrbanSound8K | LSTM–7 | Log-Mel $\rightarrow$ framewise LN $\rightarrow$ Linear($F \rightarrow 256$) $\rightarrow$ BiLSTM($H\!=\!384$, $L\!=\!2$) $\rightarrow$ Attn-pool $\rightarrow$ LN $\rightarrow$ FC. |
| UrbanSound8K | LSTM–8 | Log-Mel $\rightarrow$ framewise LN $\rightarrow$ Linear($F \rightarrow 256$) $\rightarrow$ BiLSTM($H\!=\!384$, $L\!=\!3$) $\rightarrow$ Attn-pool $\rightarrow$ LN $\rightarrow$ FC. |
| UrbanSound8K | LSTM–9 | Log-Mel $\rightarrow$ framewise LN $\rightarrow$ Linear($F \rightarrow 192$) $\rightarrow$ BiLSTM($H\!=\!384$, $L\!=\!2$) $\rightarrow$ Mean-pool $\rightarrow$ LN $\rightarrow$ FC. |

# D  MODEL POOLS ACROSS DATASETS

## D.1  TABLES: FAMILIES, COMPLEXITY, AND ACCURACY

Table 8: GLUE / SST-2: Models, MACs, and validation accuracy.

| Model | MACs | Val. Acc. |
|---|---:|---|
| *TextCNN family* | | |
| TextCNN–1 | 846,109 | 0.8085 |
| TextCNN–2 | 1,081,596 | 0.8291 |
| TextCNN–3 | 1,279,675 | 0.8050 |
| TextCNN–4 | 1,692,218 | 0.8016 |
| TextCNN–5 | 2,163,192 | 0.8108 |
| TextCNN–6 | 2,559,351 | 0.8028 |
| TextCNN–7 | 3,384,437 | 0.7936 |
| TextCNN–8 | 4,326,385 | 0.8028 |
| TextCNN–9 | 5,118,703 | 0.8245 |
| TextCNN–10 | 1,691,834 | 0.8154 |
| TextCNN–11 | 2,162,808 | 0.8142 |
| TextCNN–12 | 2,558,839 | 0.7982 |
| TextCNN–13 | 3,383,669 | 0.8360 |
| TextCNN–14 | 4,325,617 | 0.8119 |
| TextCNN–15 | 5,117,679 | 0.8119 |
| TextCNN–16 | 6,767,338 | 0.8119 |
| TextCNN–17 | 8,651,235 | 0.8073 |
| TextCNN–18 | 10,235,359 | 0.7890 |
| TextCNN–19 | 3,383,285 | 0.8165 |
| TextCNN–20 | 4,325,233 | 0.8108 |
| TextCNN–21 | 5,117,167 | 0.8303 |
| TextCNN–22 | 6,766,570 | 0.8245 |
| TextCNN–23 | 8,650,467 | 0.8234 |
| TextCNN–24 | 10,234,335 | 0.8303 |
| TextCNN–25 | 13,533,141 | 0.8050 |
| TextCNN–26 | 17,300,935 | 0.8188 |
| TextCNN–27 | 20,468,670 | 0.8222 |
| *SBERT–GCN family* | | |
| SBERT–GCN–1 | 3,395,748,032 | 0.8658 |
| SBERT–GCN–2 | 3,427,696,448 | 0.8658 |
| SBERT–GCN–3 | 3,495,322,688 | 0.8693 |
| SBERT–GCN–4 | 3,708,998,400 | 0.8567 |
| SBERT–GCN–5 | 3,772,895,232 | 0.8601 |
| SBERT–GCN–6 | 3,908,147,712 | 0.8739 |
| SBERT–GCN–7 | 6,791,496,064 | 0.8670 |
| SBERT–GCN–8 | 6,855,392,896 | 0.8670 |
| SBERT–GCN–9 | 6,990,645,376 | 0.8693 |
| SBERT–GCN–10 | 7,375,447,360 | 0.8567 |
| SBERT–GCN–11 | 7,471,292,608 | 0.8601 |
| SBERT–GCN–12 | 7,674,171,328 | 0.8750 |
| SBERT–GCN–13 | 7,998,822,336 | 0.7362 |
| SBERT–GCN–14 | 8,104,656,896 | 0.8589 |
| SBERT–GCN–15 | 8,375,161,856 | 0.8727 |

| Model | MACs | Val. Acc. |
|---|---|---|
| SBERT–GCN–16 | 13,582,992,128 | 0.8681 |
| SBERT–GCN–17 | 13,710,785,792 | 0.8704 |
| SBERT–GCN–18 | 13,981,290,752 | 0.8693 |
| SBERT–GCN–19 | 14,708,345,280 | 0.8567 |
| SBERT–GCN–20 | 14,868,087,360 | 0.8624 |
| SBERT–GCN–21 | 15,206,218,560 | 0.8739 |
| SBERT–GCN–22 | 16,060,318,080 | 0.8589 |
| SBERT–GCN–23 | 16,466,075,520 | 0.8727 |
| SBERT–GCN–24 | 18,189,192,192 | 0.8567 |
| SBERT–GCN–25 | 18,444,779,520 | 0.8612 |
| SBERT–GCN–26 | 18,985,789,440 | 0.8727 |
| SBERT–GCN–27 | 1,358,299,2128 | 0.8681 |
| *RNN–BiGRU family* | | |
| RNN–BiGRU–1 | 4,947,953 | 0.8257 |
| RNN–BiGRU–2 | 12,369,116 | 0.8498 |
| RNN–BiGRU–3 | 9,277,221 | 0.8372 |
| RNN–BiGRU–4 | 25,974,838 | 0.8349 |
| RNN–BiGRU–5 | 14,843,349 | 0.8349 |
| RNN–BiGRU–6 | 44,528,001 | 0.8440 |
| RNN–BiGRU–7 | 3,247,142 | 0.8395 |
| RNN–BiGRU–8 | 7,421,546 | 0.8326 |
| RNN–BiGRU–9 | 7,421,674 | 0.8211 |
| RNN–BiGRU–10 | 14,842,837 | 0.8383 |
| RNN–BiGRU–11 | 12,987,803 | 0.8486 |
| RNN–BiGRU–12 | 29,685,419 | 0.8475 |
| RNN–BiGRU–13 | 19,790,791 | 0.8509 |
| RNN–BiGRU–14 | 49,475,443 | 0.8647 |
| RNN–BiGRU–15 | 5,102,433 | 0.8291 |
| RNN–BiGRU–16 | 9,276,837 | 0.8234 |
| RNN–BiGRU–17 | 3,711,093 | 0.8360 |
| RNN–BiGRU–18 | 11,132,256 | 0.8406 |
| RNN–BiGRU–19 | 7,421,930 | 0.8119 |
| RNN–BiGRU–20 | 24,119,547 | 0.8521 |
| RNN–BiGRU–21 | 12,369,628 | 0.8291 |
| RNN–BiGRU–22 | 42,054,280 | 0.8303 |
| RNN–BiGRU–23 | 2,319,497 | 0.8268 |
| RNN–BiGRU–24 | 6,493,901 | 0.8383 |
| *TFIDF–LR family* | | |
| TFIDF–LR–1 | 36 | 0.7603 |
| TFIDF–LR–2 | 37 | 0.7580 |
| TFIDF–LR–3 | 37 | 0.7615 |
| TFIDF–LR–4 | 44 | 0.7752 |
| TFIDF–LR–5 | 44 | 0.7833 |
| TFIDF–LR–6 | 44 | 0.7810 |
| TFIDF–LR–7 | 47 | 0.7970 |
| TFIDF–LR–8 | 47 | 0.7821 |
| TFIDF–LR–9 | 47 | 0.7833 |
| TFIDF–LR–10 | 50 | 0.8073 |
| TFIDF–LR–11 | 49 | 0.7890 |
| TFIDF–LR–12 | 49 | 0.7890 |

| Model | MACs | Val. Acc. |
|-------|-----:|-----------|
| *BERT family* | | |
| BERT–1 | 4,753,696,180 | 0.9289 |
| BERT–2 | 4,377,380,352 | 0.9335 |
| BERT–3 | 4,377,380,352 | 0.9392 |
| BERT–4 | 4,075,744,392 | 0.9576 |
| BERT–5 | 2,188,690,944 | 0.8979 |
| BERT–6 | 553,179,246 | 0.9025 |
| BERT–7 | 4,453,142,466 | 0.9450 |
| BERT–8 | 5,129,034,606 | 0.9381 |
| *W2V–DAN family* | | |
| W2V–DAN–1 | 26,112 | 0.7592 |
| W2V–DAN–2 | 26,112 | 0.7603 |
| W2V–DAN–3 | 26,112 | 0.7615 |
| W2V–DAN–4 | 51,712 | 0.7672 |
| W2V–DAN–5 | 51,712 | 0.7626 |
| W2V–DAN–6 | 51,712 | 0.7798 |
| W2V–DAN–7 | 77,312 | 0.7718 |
| W2V–DAN–8 | 77,312 | 0.7706 |
| W2V–DAN–9 | 77,312 | 0.7741 |
| W2V–DAN–10 | 26,112 | 0.7741 |
| W2V–DAN–11 | 26,112 | 0.7718 |
| W2V–DAN–12 | 26,112 | 0.7683 |
| W2V–DAN–13 | 51,712 | 0.7752 |
| W2V–DAN–14 | 51,712 | 0.7798 |
| W2V–DAN–15 | 51,712 | 0.7856 |
| W2V–DAN–16 | 77,312 | 0.7718 |
| W2V–DAN–17 | 77,312 | 0.7752 |
| W2V–DAN–18 | 77,312 | 0.7787 |

Table 9: CIFAR-10: Models, MACs, and validation,test accuracy.

| Model | MACs | Val. Acc | Test Acc |
|-------|-----:|----------|----------|
| *HOG–SVM family* | | | |
| HOG–SVM–1 | 17,640 | 0.5136 | 0.5149 |
| HOG–SVM–2 | 52,920 | 0.4436 | 0.4371 |
| HOG–SVM–3 | 3,240 | 0.5118 | 0.5129 |
| HOG–SVM–4 | 9,720 | 0.5248 | 0.5278 |
| *SimpleCNN family* | | | |
| SimpleCNN-1 | 5,000,000 | 0.8418 | 0.8435 |
| SimpleCNN-2 | 19,441,280 | 0.8802 | 0.8783 |
| SimpleCNN-3 | 43,330,304 | 0.8998 | 0.8955 |
| SimpleCNN-4 | 76,614,912 | 0.9076 | 0.9032 |
| SimpleCNN-5 | 76,647,680 | 0.9036 | 0.9020 |
| SimpleCNN-6 | 76,749,824 | 0.9050 | 0.9028 |
| SimpleCNN-7 | 171,644,416 | 0.9196 | 0.9104 |
| SimpleCNN-8 | 304,355,840 | 0.9132 | 0.9109 |

| Model | MACs | Val. Acc | Test Acc |
|---|---:|---|---|
| *MLP family* | | | |
| MLP-1 | 15,360 | 0.3510 | 0.3540 |
| MLP-2 | 410,240 | 0.5352 | 0.5265 |
| MLP-3 | 853,248 | 0.5206 | 0.5173 |
| MLP-4 | 1,837,568 | 0.5368 | 0.5310 |
| MLP-5 | 1,901,824 | 0.5068 | 0.5137 |
| MLP-6 | 1,917,568 | 0.3828 | 0.3924 |
| MLP-7 | 3,048,704 | 0.4488 | 0.4493 |
| MLP-8 | 4,199,424 | 0.4938 | 0.4937 |
| *ResNet family* | | | |
| ResNet-1 | 1,824,805,898 | 0.9292 | 0.9232 |
| ResNet-2 | 3,679,511,562 | 0.9224 | 0.9174 |
| ResNet-3 | 4,130,408,458 | 0.8900 | 0.8876 |
| ResNet-4 | 11,600,611,338 | 0.8902 | 0.8854 |
| ResNet-5 | 4,284,850,186 | 0.9110 | 0.9063 |
| ResNet-6 | 16,535,621,642 | 0.9210 | 0.9151 |
| ResNet-7 | 22,834,716,682 | 0.8966 | 0.8904 |
| *MobileNetV2 family* | | | |
| MobileNetV2-1 | 31,903,440 | 0.8660 | 0.8588 |
| MobileNetV2-2 | 51,603,792 | 0.8814 | 0.8755 |
| MobileNetV2-3 | 109,787,960 | 0.8990 | 0.9017 |
| MobileNetV2-4 | 156,463,008 | 0.9054 | 0.9017 |
| MobileNetV2-5 | 223,366,824 | 0.9140 | 0.9055 |
| MobileNetV2-6 | 299,828,200 | 0.9070 | 0.9041 |
| *ShuffleNetV2 family* | | | |
| ShuffleNetV2-1 | 20,755,248 | 0.8830 | 0.8796 |
| ShuffleNetV2-2 | 73,897,316 | 0.9088 | 0.9017 |
| ShuffleNetV2-3 | 150,140,336 | 0.9120 | 0.9111 |
| ShuffleNetV2-4 | 294,356,964 | 0.9170 | 0.9119 |
| *GhostNet family* | | | |
| GhostNet-1 | 22,311,332 | 0.8972 | 0.8931 |
| GhostNet-2 | 73,503,368 | 0.9152 | 0.9096 |
| GhostNet-3 | 117,486,180 | 0.9146 | 0.9154 |
| GhostNet-4 | 88,153,336 | 0.9150 | 0.9128 |
| GhostNet-5 | 140,844,508 | 0.9168 | 0.9211 |
| GhostNet-6 | 207,043,596 | 0.9234 | 0.9193 |
| *DeiT family* | | | |
| DeiT-1 | 1,074,661,248 | 0.7618 | 0.7606 |
| DeiT-2 | 4,240,838,400 | 0.7608 | 0.7563 |
| DeiT-3 | 16,847,740,416 | 0.6596 | 0.6521 |
| *Swin family* | | | |
| Swin-1 | 4,371,091,488 | 0.7774 | 0.7851 |
| Swin-2 | 8,544,235,680 | 0.7706 | 0.7693 |
| *ConvNeXt family* | | | |
| ConvNeXt-1 | 4,454,809,344 | 0.8888 | 0.8851 |
| ConvNeXt-2 | 8,682,990,336 | 0.8938 | 0.8915 |
| ConvNeXt-3 | 15,353,766,912 | 0.8980 | 0.8969 |

Table 10: UrbanSound8K: Models, MACs, and validation,test accuracy.

| Model | MACs | Val. Acc. | Test Acc. |
|---|---:|---|---|
| *TinyCNN family* | | | |
| TinyCNN-1 | 256,000,000 | 0.6899 | 0.7121 |
| TinyCNN-2 | 963,000,000 | 0.7818 | 0.8005 |
| TinyCNN-3 | 841,000,000 | 0.7633 | 0.8172 |
| TinyCNN-4 | 3,230,000,000 | 0.7540 | 0.8148 |
| TinyCNN-5 | 153,000,000 | 0.7497 | 0.8172 |
| TinyCNN-6 | 2,153,000,000 | 0.7571 | 0.7957 |
| TinyCNN-7 | 1,926,000,000 | 0.7848 | 0.7861 |
| TinyCNN-8 | 963,000,000 | 0.7534 | 0.7670 |
| *CRNN family* | | | |
| CRNN-1 | 973,000,000 | 0.7515 | 0.8124 |
| CRNN-2 | 7,985,000,000 | 0.7552 | 0.8602 |
| CRNN-3 | 117,000,000 | 0.7158 | 0.5663 |
| CRNN-4 | 3,324,000,000 | 0.7577 | 0.8256 |
| CRNN-5 | 156,000,000 | 0.7497 | 0.6989 |
| CRNN-6 | 988,000,000 | 0.7707 | 0.8124 |
| CRNN-7 | 1,946,000,000 | 0.7546 | 0.8029 |
| CRNN-8 | 850,000,000 | 0.7756 | 0.8088 |
| CRNN-9 | 2,188,000,000 | 0.7571 | 0.8315 |
| *ResNet family* | | | |
| ResNet-1 | 899,000,000 | 0.7633 | 0.7145 |
| ResNet-2 | 5,525,000,000 | 0.7670 | 0.8148 |
| ResNet-3 | 7,872,000,000 | 0.7633 | 0.8041 |
| ResNet-4 | 1,850,000,000 | 0.7651 | 0.7348 |
| ResNet-5 | 11,432,000,000 | 0.7737 | 0.7993 |
| ResNet-6 | 16,380,000,000 | 0.7633 | 0.7957 |
| ResNet-7 | 2,082,000,000 | 0.7707 | 0.7431 |
| ResNet-8 | 12,858,000,000 | 0.7626 | 0.7168 |
| ResNet-9 | 18,501,000,000 | 0.7552 | 0.7957 |
| *PANN family* | | | |
| PANN-1 | 957,000,000 | 0.5808 | 0.6260 |
| PANN-2 | 1,489,000,000 | 0.6011 | 0.6069 |
| PANN-3 | 542,000,000 | 0.5031 | 0.4624 |
| PANN-4 | 1,418,000,000 | 0.6369 | 0.6177 |
| PANN-5 | 2,209,000,000 | 0.6492 | 0.6870 |
| PANN-6 | 801,000,000 | 0.5425 | 0.5818 |
| *AST family* | | | |
| AST-1 | 1,487,000,000 | 0.6634 | 0.6906 |
| AST-2 | 3,087,000,000 | 0.7004 | 0.7121 |
| AST-3 | 749,000,000 | 0.7102 | 0.6977 |
| AST-4 | 733,000,000 | 0.7010 | 0.6894 |
| AST-5 | 293,000,000 | 0.6702 | 0.7073 |
| AST-6 | 628,000,000 | 0.6763 | 0.7312 |
| AST-7 | 147,000,000 | 0.6634 | 0.6571 |
| AST-8 | 143,000,000 | 0.6887 | 0.6643 |
| *LSTM family* | | | |

| Model | MACs | Val. Acc. | Test Acc. |
|---|---|---|---|
| LSTM-1 | 996,000,000 | 0.6301 | 0.5747 |
| LSTM-2 | 1,050,000,000 | 0.4605 | 0.4229 |
| LSTM-3 | 1,622,000,000 | 0.6578 | 0.6320 |
| LSTM-4 | 2,198,000,000 | 0.6449 | 0.5771 |
| LSTM-5 | 3,606,000,000 | 0.6609 | 0.5974 |
| LSTM-6 | 2,118,000,000 | 0.6406 | 0.5281 |
| LSTM-7 | 108,000,000 | 0.4470 | 0.3967 |
| LSTM-8 | 264,000,000 | 0.6091 | 0.5484 |
| LSTM-9 | 292,000,000 | 0.5919 | 0.5484 |

## D.2 FIGURES: $\varepsilon$-PARETO FRONTIERS

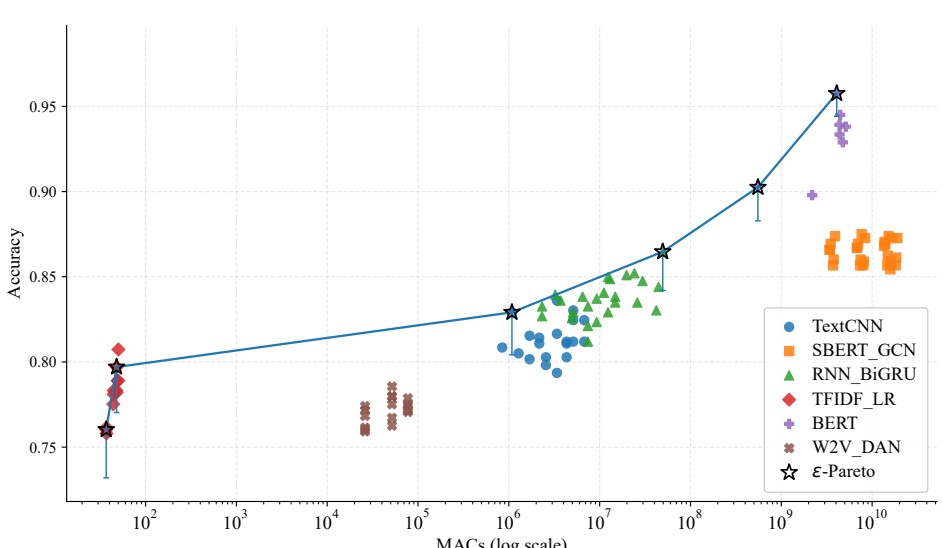

Figure 5: $\varepsilon$-Pareto frontier on GLUE/SST-2.

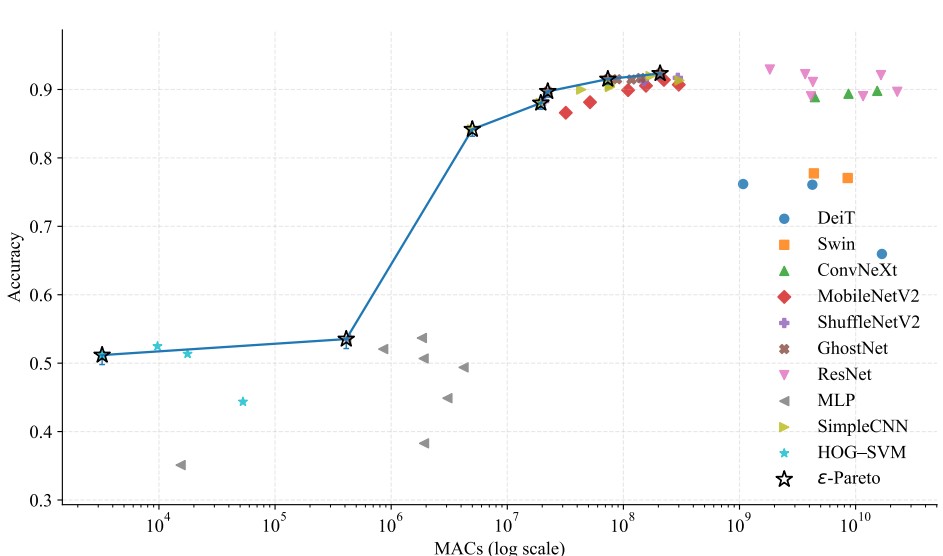

Figure 6: $\varepsilon$-Pareto frontier on CIFAR-10.

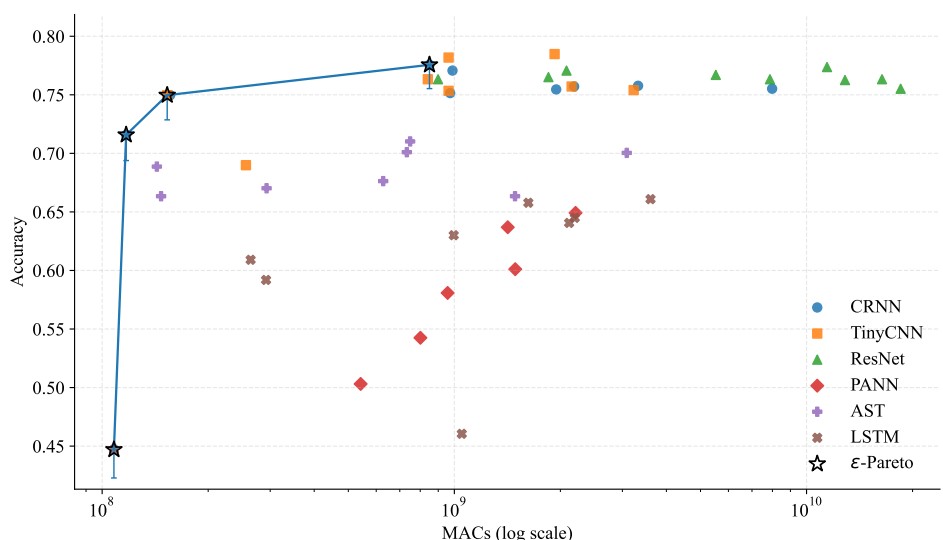

Figure 7: $\varepsilon$-Pareto frontier on UrbanSound8K.

# E CASCADE SELECTION RESULTS

## E.1 TABLES: STABILITY & COMPUTE RANKINGS

Table 11: Accuracy tolerance = 0.03, Final cascades on GLUE / SST-2: StabIndex and MACs

| Cascade | K | MACs | StabIndex | Rank(MACs) | Rank(Stab.) | Rank Sum |
|---|---|---|---|---|---|---|
| TFIDF-LR-3 → BERT-4 → BERT-6 | 3 | 1027581945 | 0.016485 | 2 | 1 | 3 |
| BERT-4 → BERT-6 | 2 | 1146779634 | 0.016765 | 4 | 2 | 6 |
| TFIDF-LR-1 → TFIDF-LR-3 → RNN-BiGRU-24 → BERT-6 | 4 | 1867030971 | 0.017915 | 5 | 3 | 8 |
| TFIDF-LR-3 → RNN-BiGRU-24 → BERT-6 | 3 | 1867144427 | 0.018777 | 6 | 4 | 10 |
| TFIDF-LR-1 → TFIDF-LR-3 → BERT-4 → BERT-6 | 4 | 1006139180 | 0.020207 | 1 | 9 | 10 |
| TFIDF-LR-1 → BERT-4 → BERT-6 | 3 | 1048424236 | 0.020207 | 7 | 9 | 12 |
| TFIDF-LR-1 → TFIDF-LR-3 → TextCNN-1 → BERT-4 | 4 | 2427089429 | 0.019064 | 10 | 5 | 15 |
| RNN-BiGRU-24 → BERT-6 | 2 | 1942515072 | 0.020182 | 8 | 6 | 14 |
| TFIDF-LR-1 → BERT-6 | 2 | 2747859968 | 0.019064 | 13 | 5 | 18 |
| TFIDF-LR-3 → TextCNN-1 → BERT-6 | 3 | 2551900160 | 0.019922 | 12 | 7 | 19 |
| TFIDF-LR-1 → TFIDF-LR-3 → RNN-BiGRU-24 → BERT-6 | 4 | 2418025728 | 0.020205 | 9 | 11 | 20 |
| TFIDF-LR-1 → RNN-BiGRU-24 → BERT-6 | 3 | 1942513920 | 0.021638 | 7 | 15 | 22 |
| TFIDF-LR-1 → TFIDF-LR-3 → BERT-6 | 3 | 2551900160 | 0.021356 | 11 | 14 | 25 |
| TextCNN-1 → BERT-6 | 2 | 4069830656 | 0.021042 | 15 | 12 | 27 |
| TFIDF-LR-1 → TextCNN-1 → BERT-6 | 3 | 2747859968 | 0.021063 | 14 | 13 | 27 |

Table 12: Accuracy tolerance = 0.05, Final cascades on GLUE / SST-2: StabIndex and MACs

| Cascade | K | MACs | StabIndex | Rank(MACs) | Rank(Stab.) | Rank Sum |
|---|---|---|---|---|---|---|
| TFIDF-LR-1 → TFIDF-LR-3 → BERT-4 → BERT-6 | 4 | 610,749,160 | 0.01230 | 4 | 2 | 6 |
| TFIDF-LR-1 → BERT-4 → BERT-6 | 3 | 671,219,151 | 0.01112 | 8 | 1 | 9 |
| TFIDF-LR-3 → BERT-4 → BERT-6 | 3 | 668,382,576 | 0.01275 | 7 | 3 | 10 |
| TFIDF-LR-3 → TextCNN-1 → BERT-4 → BERT-6 | 4 | 643,700,819 | 0.01455 | 6 | 5 | 11 |

| Cascade | K | MACs | StabIndex | Rank(MACs) | Rank(Stab.) | Rank Sum |
|---|---|---|---|---|---|---|
| TFIDF-LR-3 → RNN-BiGRU-24 → BERT-4 → BERT-6 | 4 | 601,132,160 | 0.01939 | 3 | 15 | 18 |
| TFIDF-LR-1 → TFIDF-LR-3 → RNN-BiGRU-24 → BERT-6 | 4 | 1,278,466,538 | 0.01639 | 13 | 6 | 19 |
| BERT-4 → BERT-6 | 2 | 828,971,272 | 0.01842 | 9 | 12 | 21 |
| TFIDF-LR-1 → TFIDF-LR-3 → RNN-BiGRU-24 → BERT-4 | 4 | 600,777,008 | 0.02168 | 2 | 20 | 22 |
| TFIDF-LR-1 → TextCNN-1 → RNN-BiGRU-24 → BERT-6 | 4 | 1,191,062,874 | 0.01726 | 12 | 10 | 22 |
| TFIDF-LR-3 → TextCNN-1 → BERT-6 | 3 | 1,543,614,191 | 0.01383 | 19 | 4 | 23 |
| TFIDF-LR-1 → TextCNN-1 → RNN-BiGRU-24 → BERT-6 | 5 | 1,151,012,706 | 0.01869 | 10 | 13 | 23 |
| TFIDF-LR-3 → RNN-BiGRU-24 → BERT-6 | 3 | 1,307,702,130 | 0.01680 | 16 | 8 | 24 |
| TFIDF-LR-1 → TextCNN-1 → BERT-6 | 3 | 1,580,582,441 | 0.01658 | 20 | 7 | 27 |
| TFIDF-LR-1 → TextCNN-1 → BERT-4 → BERT-6 | 4 | 614,349,903 | 0.02374 | 5 | 22 | 27 |
| TFIDF-LR-1 → TFIDF-LR-3 → TextCNN-1 → BERT-6 | 4 | 1,482,239,529 | 0.01683 | 18 | 9 | 27 |
| TFIDF-LR-1 → RNN-BiGRU-24 → BERT-6 | 3 | 1,305,956,967 | 0.02023 | 15 | 16 | 31 |
| TextCNN-1 → BERT-6 | 2 | 1,823,948,928 | 0.01757 | 21 | 11 | 32 |
| TextCNN-1 → RNN-BiGRU-24 → BERT-6 | 3 | 1,282,201,057 | 0.02098 | 14 | 18 | 32 |
| RNN-BiGRU-24 → BERT-6 | 2 | 1,442,333,045 | 0.02039 | 17 | 17 | 34 |
| TFIDF-LR-1 → TextCNN-1 → RNN-BiGRU-24 → BERT-6 | 4 | 1,156,046,979 | 0.02443 | 11 | 23 | 34 |
| TFIDF-LR-3 → BERT-6 | 2 | 1,897,651,659 | 0.01898 | 23 | 14 | 37 |
| TFIDF-LR-1 → TFIDF-LR-3 → BERT-6 | 3 | 1,860,259,522 | 0.02100 | 22 | 19 | 41 |
| TFIDF-LR-1 → BERT-6 | 2 | 2,033,198,201 | 0.02771 | 24 | 24 | 48 |

Table 13: Accuracy tolerance = 0.08, Final cascades on GLUE / SST-2: StabIndex and MACs

| Cascade | K | MACs | StabIndex | Rank(MACs) | Rank(Stab.) | Rank Sum |
|---|---|---|---|---|---|---|
| TFIDF-LR-1 → TFIDF-LR-3 → RNN-BiGRU-24 → BERT-4 | 4 | 207,765,075 | 0.01450 | 4 | 4 | 8 |
| TFIDF-LR-3 → RNN-BiGRU-24 → BERT-4 | 3 | 219,132,579 | 0.02037 | 5 | 8 | 13 |
| TFIDF-LR-3 → BERT-4 | 2 | 406,637,523 | 0.01500 | 14 | 5 | 19 |
| TFIDF-LR-1 → TFIDF-LR-3 → BERT-4 | 3 | 393,315,563 | 0.01857 | 12 | 7 | 19 |

| Cascade | K | MACs | StabIndex | Rank(MACs) | Rank(Stab.) | Rank Sum |
|---|---|---|---|---|---|---|
| RNN-BiGRU-24 → BERT-4 | 2 | 238,520,644 | 0.02308 | 7 | 12 | 19 |
| TFIDF-LR-1 → RNN-BiGRU-24 → BERT-4 | 3 | 220,189,487 | 0.02309 | 6 | 13 | 19 |
| TFIDF-LR-1 → TextCNN-1 → RNN-BiGRU-24 → BERT-4 | 4 | 196,125,682 | 0.02470 | 2 | 19 | 21 |
| TFIDF-LR-1 → TFIDF-LR-3 → TextCNN-1 → RNN-BiGRU-24 → BERT-4 | 5 | 196,061,526 | 0.02577 | 1 | 22 | 23 |
| TFIDF-LR-1 → TFIDF-LR-3 → RNN-BiGRU-24 → BERT-6 | 4 | 546,355,518 | 0.01423 | 22 | 2 | 24 |
| TFIDF-LR-1 → TFIDF-LR-3 → TextCNN-1 → BERT-4 | 4 | 394,084,588 | 0.02286 | 13 | 11 | 24 |
| TFIDF-LR-3 → RNN-BiGRU-24 → BERT-6 | 3 | 555,185,103 | 0.01308 | 24 | 1 | 25 |
| TFIDF-LR-1 → TextCNN-1 → BERT-4 | 3 | 310,538,681 | 0.02324 | 11 | 14 | 25 |
| TFIDF-LR-1 → TFIDF-LR-3 → BERT-4 → BERT-6 | 4 | 270,880,246 | 0.02342 | 10 | 15 | 25 |
| TFIDF-LR-1 → RNN-BiGRU-24 → BERT-6 | 3 | 516,374,502 | 0.01554 | 20 | 6 | 26 |
| RNN-BiGRU-24 → BERT-6 | 2 | 605,683,680 | 0.01438 | 25 | 3 | 28 |
| TextCNN-1 → RNN-BiGRU-24 → BERT-6 | 3 | 523,067,208 | 0.02140 | 21 | 10 | 31 |
| TFIDF-LR-3 → TextCNN-1 → BERT-4 | 3 | 407,432,596 | 0.02431 | 15 | 17 | 32 |
| TFIDF-LR-1 → TFIDF-LR-3 → TextCNN-1 → BERT-4 → BERT-6 | 5 | 199,287,673 | 0.03143 | 3 | 30 | 33 |
| TFIDF-LR-1 → BERT-4 | 2 | 430,743,955 | 0.02597 | 16 | 23 | 39 |
| TFIDF-LR-3 → TextCNN-1 → BERT-4 → BERT-6 | 4 | 251,083,240 | 0.03373 | 8 | 31 | 39 |
| TFIDF-LR-1 → BERT-6 | 2 | 1,411,553,700 | 0.02130 | 32 | 9 | 41 |
| TextCNN-1 → BERT-4 | 2 | 553,626,462 | 0.02467 | 23 | 18 | 41 |
| TFIDF-LR-1 → TextCNN-1 → BERT-4 → BERT-6 | 4 | 255,012,483 | 0.03862 | 9 | 32 | 41 |
| TFIDF-LR-3 → TextCNN-1 → RNN-BiGRU-24 → BERT-6 | 4 | 475,176,130 | 0.02683 | 18 | 25 | 43 |
| TFIDF-LR-1 → TextCNN-1 → RNN-BiGRU-24 → BERT-6 | 4 | 470,051,288 | 0.02797 | 17 | 27 | 44 |
| TFIDF-LR-1 → TFIDF-LR-3 → TextCNN-1 → RNN-BiGRU-24 → BERT-6 | 5 | 475,176,151 | 0.02797 | 19 | 27 | 46 |
| TFIDF-LR-3 → BERT-6 | 2 | 1,126,438,557 | 0.02417 | 31 | 16 | 47 |
| TFIDF-LR-3 → TextCNN-1 → BERT-6 | 3 | 799,791,838 | 0.02531 | 27 | 20 | 47 |
| TFIDF-LR-1 → TFIDF-LR-3 → BERT-6 | 3 | 1,121,764,551 | 0.02571 | 30 | 21 | 51 |

| Cascade | K | MACs | StabIndex | Rank(MACs) | Rank(Stab.) | Rank Sum |
|---|---|---|---|---|---|---|
| TFIDF-LR-1 → TFIDF-LR-3 → TextCNN-1 → BERT-6 | 4 | 790,436,372 | 0.02732 | 26 | 26 | 52 |
| TextCNN-1 → BERT-6 | 2 | 991,973,581 | 0.02648 | 29 | 24 | 53 |
| TFIDF-LR-1 → TextCNN-1 → BERT-6 | 3 | 870,095,613 | 0.02935 | 28 | 29 | 57 |

Table 14: Accuracy tolerance = 0.03, Final cascades on CIFAR-10: StabIndex, MACs, and cascade depth $K$.

| Cascade | $K$ | MACs | StabIndex | Rank(MACs) | Rank(Stab.) | Rank Sum |
|---|---|---|---|---|---|---|
| SimpleCNN-1 → GhostNet-2 | 2 | 49,499,918 | 0.00385 | 18 | 1 | 19 |
| HOG–SVM–3 → SimpleCNN-1 → SimpleCNN-2 → GhostNet-1 | 4 | 41,146,867 | 0.00708 | 7 | 13 | 20 |
| MLP-2 → SimpleCNN-1 → GhostNet-2 | 3 | 50,165,044 | 0.00395 | 20 | 2 | 22 |
| MLP-2 → SimpleCNN-1 → GhostNet-2 → GhostNet-6 | 4 | 49,721,086 | 0.00415 | 19 | 3 | 22 |
| SimpleCNN-1 → SimpleCNN-2 → GhostNet-1 | 3 | 41,313,283 | 0.00717 | 8 | 15 | 23 |
| SimpleCNN-1 → GhostNet-2 | 2 | 42,146,321 | 0.00745 | 9 | 15 | 24 |
| HOG–SVM–3 → SimpleCNN-1 → GhostNet-2 | 3 | 42,045,794 | 0.00765 | 8 | 17 | 25 |
| MLP-2 → SimpleCNN-2 → GhostNet-1 → GhostNet-2 | 4 | 43,472,508 | 0.00763 | 12 | 17 | 29 |
| HOG–SVM–3 → SimpleCNN-1 → GhostNet-1 → GhostNet-2 | 4 | 45,161,061 | 0.00744 | 14 | 15 | 29 |
| SimpleCNN-2 → GhostNet-6 | 2 | 46,323,879 | 0.00915 | 28 | 25 | 53 |
| MLP-2 → SimpleCNN-1 → SimpleCNN-2 → GhostNet-6 | 4 | 67,857,275 | 0.00800 | 35 | 22 | 57 |
| MLP-2 → GhostNet-1 → GhostNet-6 | 3 | 56,777,797 | 0.00805 | 32 | 26 | 58 |
| MLP-2 → SimpleCNN-2 → GhostNet-2 | 3 | 52,737,284 | 0.00830 | 26 | 32 | 58 |
| MLP-2 → SimpleCNN-2 → GhostNet-2 → GhostNet-6 | 4 | 52,737,284 | 0.00830 | 26 | 32 | 58 |
| SimpleCNN-1 → SimpleCNN-2 → GhostNet-6 | 3 | 67,177,473 | 0.00820 | 34 | 29 | 63 |
| HOG–SVM–3 → SimpleCNN-2 → GhostNet-2 → GhostNet-6 | 4 | 53,292,130 | 0.00965 | 28 | 42 | 70 |
| HOG–SVM–3 → SimpleCNN-2 → GhostNet-2 | 3 | 53,712,528 | 0.01000 | 29 | 44 | 73 |
| SimpleCNN-2 → GhostNet-2 | 2 | 54,109,424 | 0.00985 | 31 | 43 | 74 |
| HOG–SVM–3 → MLP-2 → GhostNet-2 | 3 | 124,949,502 | 0.00890 | 42 | 39 | 81 |
| HOG–SVM–3 → GhostNet-6 | 2 | 356,264,185 | 0.00940 | 47 | 41 | 88 |

| Cascade | $K$ | MACs | StabIndex | Rank(MACs) | Rank(Stab.) | Rank Sum |
|---|---|---|---|---|---|---|
| MLP-2 → GhostNet-2 | 2 | 127,213,581 | 0.01030 | 43 | 45 | 88 |
| MLP-2 → GhostNet-6 | 2 | 334,953,529 | 0.01070 | 46 | 46 | 92 |
| HOG–SVM–3 → MLP-2 → GhostNet-6 | 3 | 327,051,349 | 0.01095 | 45 | 47 | 92 |

Table 15: Accuracy tolerance = 0.05, Final cascades on CIFAR-10: StabIndex, MACs, and cascade depth $K$.

| Cascade | $K$ | MACs | StabIndex | Rank(MACs) | Rank(Stab.) | Rank Sum |
|---|---|---|---|---|---|---|
| SimpleCNN-1 → GhostNet-1 | 2 | 18,460,828 | 0.00385 | 1 | 1 | 2 |
| MLP-2 → SimpleCNN-1 → GhostNet-1 | 3 | 19,156,321 | 0.00435 | 3 | 4 | 7 |
| HOG–SVM–3 → MLP-2 → SimpleCNN-1 → GhostNet-1 | 4 | 19,065,833 | 0.00470 | 2 | 5 | 7 |
| SimpleCNN-1 → GhostNet-2 | 2 | 30,610,974 | 0.00410 | 10 | 2 | 12 |
| HOG–SVM–3 → SimpleCNN-1 → GhostNet-2 | 3 | 30,510,447 | 0.00430 | 9 | 3 | 12 |
| MLP-2 → SimpleCNN-1 → SimpleCNN-2 → GhostNet-2 | 4 | 24,737,094 | 0.00585 | 8 | 7 | 15 |
| HOG–SVM–3 → SimpleCNN-1 → SimpleCNN-2 → GhostNet-2 | 4 | 24,189,871 | 0.00635 | 6 | 9 | 15 |
| MLP-2 → SimpleCNN-1 → SimpleCNN-2 → GhostNet-1 | 4 | 22,102,659 | 0.00645 | 5 | 10 | 15 |
| MLP-2 → SimpleCNN-1 → GhostNet-2 | 3 | 30,705,121 | 0.00495 | 11 | 6 | 17 |
| SimpleCNN-1 → SimpleCNN-2 → GhostNet-2 | 3 | 24,290,398 | 0.00775 | 7 | 12 | 19 |
| SimpleCNN-1 → SimpleCNN-2 → GhostNet-1 | 3 | 21,620,990 | 0.00895 | 4 | 23 | 27 |
| HOG–SVM–3 → MLP-2 → SimpleCNN-2 → GhostNet-2 | 4 | 39,447,111 | 0.00815 | 19 | 13 | 32 |
| HOG–SVM–3 → SimpleCNN-2 → GhostNet-2 | 3 | 40,892,964 | 0.00815 | 22 | 13 | 35 |
| HOG–SVM–3 → SimpleCNN-2 → GhostNet-1 | 3 | 39,095,094 | 0.00855 | 18 | 18 | 36 |
| SimpleCNN-1 → SimpleCNN-2 → GhostNet-6 | 3 | 30,809,466 | 0.00910 | 12 | 24 | 36 |
| MLP-2 → SimpleCNN-2 → GhostNet-2 | 3 | 40,000,978 | 0.00870 | 20 | 20 | 40 |
| MLP-2 → SimpleCNN-2 → GhostNet-6 | 3 | 44,211,186 | 0.00830 | 26 | 15 | 41 |
| HOG–SVM–3 → MLP-2 → SimpleCNN-2 → GhostNet-6 | 4 | 43,884,367 | 0.00840 | 25 | 16 | 41 |
| HOG–SVM–3 → GhostNet-2 | 2 | 121,111,133 | 0.00630 | 35 | 8 | 43 |
| SimpleCNN-1 → GhostNet-6 | 2 | 62,539,732 | 0.00760 | 32 | 11 | 43 |

| Cascade | $K$ | MACs | StabIndex | Rank(MACs) | Rank(Stab.) | Rank Sum |
|---|---|---|---|---|---|---|
| MLP-2 → SimpleCNN-1 → SimpleCNN-2 → GhostNet-6 | 4 | 31,497,114 | 0.01015 | 13 | 30 | 43 |
| HOG–SVM–3 → SimpleCNN-2 → GhostNet-6 | 3 | 45,042,247 | 0.00845 | 27 | 17 | 44 |
| MLP-2 → GhostNet-1 | 2 | 38,985,854 | 0.01005 | 17 | 29 | 46 |
| HOG–SVM–3 → MLP-2 → GhostNet-1 | 3 | 38,383,632 | 0.01025 | 16 | 31 | 47 |
| HOG–SVM–3 → MLP-2 → SimpleCNN-2 → GhostNet-1 | 4 | 37,111,825 | 0.01145 | 14 | 34 | 48 |
| SimpleCNN-2 → GhostNet-1 | 2 | 40,019,254 | 0.00965 | 21 | 28 | 49 |
| HOG–SVM–3 → SimpleCNN-1 → GhostNet-6 | 3 | 62,147,208 | 0.00860 | 31 | 19 | 50 |
| SimpleCNN-2 → GhostNet-2 | 2 | 41,779,985 | 0.00925 | 24 | 26 | 50 |
| MLP-2 → SimpleCNN-2 → GhostNet-1 | 3 | 37,410,323 | 0.01190 | 15 | 35 | 50 |
| MLP-2 → SimpleCNN-1 → GhostNet-6 | 3 | 61,924,718 | 0.00880 | 30 | 21 | 51 |
| HOG–SVM–3 → MLP-2 → SimpleCNN-1 → GhostNet-6 | 4 | 61,836,156 | 0.00890 | 29 | 22 | 51 |
| SimpleCNN-2 → GhostNet-6 | 2 | 46,323,879 | 0.00915 | 28 | 25 | 53 |
| HOG–SVM–3 → GhostNet-1 | 2 | 41,685,671 | 0.01140 | 23 | 33 | 56 |
| HOG–SVM–3 → MLP-2 → GhostNet-2 | 3 | 111,101,265 | 0.00960 | 33 | 27 | 60 |
| HOG–SVM–3 → GhostNet-6 | 2 | 322,225,978 | 0.01090 | 38 | 32 | 70 |
| MLP-2 → GhostNet-2 | 2 | 114,762,669 | 0.01370 | 34 | 36 | 70 |
| MLP-2 → GhostNet-6 | 2 | 304,631,652 | 0.01410 | 37 | 37 | 74 |
| HOG–SVM–3 → MLP-2 → GhostNet-6 | 3 | 295,378,079 | 0.01445 | 36 | 38 | 74 |

Table 16: Accuracy tolerance = 0.08, Final cascades on CIFAR-10: StabIndex, MACs, and cascade depth $K$.

| Cascade | $K$ | MACs | StabIndex | Rank(MACs) | Rank(Stab.) | Rank Sum |
|---|---|---|---|---|---|---|
| SimpleCNN-1 → GhostNet-1 | 2 | 11,650,158 | 0.00715 | 2 | 5 | 7 |
| HOG–SVM–3 → SimpleCNN-1 → GhostNet-1 | 3 | 11,549,630 | 0.00775 | 1 | 7 | 8 |
| SimpleCNN-1 → GhostNet-2 | 2 | 14,941,046 | 0.00655 | 8 | 2 | 10 |
| HOG–SVM–3 → SimpleCNN-1 → GhostNet-2 | 3 | 14,840,519 | 0.00710 | 7 | 4 | 11 |
| MLP-2 → SimpleCNN-1 → GhostNet-2 | 3 | 15,156,665 | 0.00795 | 10 | 9 | 19 |
| HOG–SVM–3 → MLP-2 → SimpleCNN-1 → GhostNet-2 | 4 | 15,062,000 | 0.00815 | 9 | 10 | 19 |
| HOG–SVM–3 → SimpleCNN-1 → SimpleCNN-2 | 3 | 12,089,105 | 0.00925 | 4 | 16 | 20 |

| Cascade | $K$ | MACs | StabIndex | Rank(MACs) | Rank(Stab.) | Rank Sum |
|---|---|---|---|---|---|---|
| MLP-2 → SimpleCNN-1 → GhostNet-1 | 3 | 11,692,026 | 0.00940 | 3 | 17 | 20 |
| HOG–SVM–3 → SimpleCNN-2 | 2 | 31,794,475 | 0.00700 | 18 | 3 | 21 |
| SimpleCNN-1 → GhostNet-6 | 2 | 23,433,801 | 0.00785 | 14 | 8 | 22 |
| HOG–SVM–3 → MLP-2 → GhostNet-2 | 3 | 93,443,576 | 0.00360 | 22 | 1 | 23 |
| SimpleCNN-1 → SimpleCNN-2 | 2 | 12,189,632 | 0.00965 | 5 | 18 | 23 |
| HOG–SVM–3 → SimpleCNN-1 → GhostNet-6 | 3 | 23,125,739 | 0.00830 | 13 | 12 | 25 |
| MLP-2 → SimpleCNN-1 → GhostNet-6 | 3 | 22,849,402 | 0.00900 | 12 | 14 | 26 |
| HOG–SVM–3 → MLP-2 → SimpleCNN-1 → GhostNet-6 | 4 | 22,777,118 | 0.00925 | 11 | 15 | 26 |
| MLP-2 → GhostNet-2 | 2 | 97,118,816 | 0.00770 | 23 | 6 | 29 |
| MLP-2 → SimpleCNN-1 → SimpleCNN-2 | 3 | 12,296,143 | 0.01195 | 6 | 25 | 31 |
| HOG–SVM–3 → GhostNet-2 | 2 | 103,133,232 | 0.00820 | 24 | 11 | 35 |
| HOG–SVM–3 → MLP-2 → SimpleCNN-2 | 3 | 29,232,834 | 0.01125 | 15 | 21 | 36 |
| HOG–SVM–3 → MLP-2 → SimpleCNN-2 → GhostNet-1 | 4 | 29,232,834 | 0.01130 | 15 | 22 | 37 |
| MLP-2 → GhostNet-6 | 2 | 264,258,790 | 0.00890 | 26 | 13 | 39 |
| MLP-2 → GhostNet-1 | 2 | 32,711,383 | 0.01005 | 20 | 19 | 39 |
| HOG–SVM–3 → MLP-2 → GhostNet-1 | 3 | 31,850,484 | 0.01050 | 19 | 20 | 39 |
| MLP-2 → SimpleCNN-2 | 2 | 30,259,133 | 0.01460 | 17 | 26 | 43 |
| HOG–SVM–3 → GhostNet-1 | 2 | 34,715,082 | 0.01190 | 21 | 24 | 45 |
| HOG–SVM–3 → MLP-2 → GhostNet-6 | 3 | 251,605,862 | 0.01180 | 25 | 23 | 48 |
| HOG–SVM–3 → GhostNet-6 | 2 | 275,602,924 | 0.01465 | 27 | 27 | 54 |

Table 17: Accuracy tolerance = 0.03, Final cascades on UrbanSound8K: StabIndex, MACs, and cascade depth $K$.

| Cascade | $K$ | MACs | StabIndex | Rank(MACs) | Rank(Stab.) | Rank Sum |
|---|---|---|---|---|---|---|
| TinyCNN-5 → CRNN-8 | 2 | 386,934,464 | 0.01871 | 2 | 5 | 7 |
| CRNN-3 → CRNN-8 | 2 | 538,162,560 | 0.01580 | 5 | 2 | 7 |
| LSTM-7 → CRNN-3 → CRNN-8 | 3 | 641,538,048 | 0.01534 | 6 | 1 | 7 |
| CRNN-3 → TinyCNN-5 → CRNN-8 | 3 | 386,889,216 | 0.02196 | 1 | 7 | 8 |
| LSTM-7 → CRNN-3 → TinyCNN-5 → CRNN-8 | 4 | 492,814,336 | 0.01821 | 4 | 4 | 8 |

Table 18: Accuracy tolerance = 0.05, Final cascades on UrbanSound8K: StabIndex, MACs, and cascade depth $K$.

| Cascade | $K$ | MACs | StabIndex | Rank(MACs) | Rank(Stab.) | Rank Sum |
|---|---|---|---|---|---|---|
| CRNN-3 → CRNN-8 | 2 | 306,278,592 | 0.01404 | 3 | 2 | 5 |
| TinyCNN-5 → CRNN-8 | 2 | 286,520,512 | 0.01793 | 2 | 5 | 7 |
| CRNN-3 → TinyCNN-5 → CRNN-8 | 3 | 256,190,912 | 0.02128 | 1 | 6 | 7 |
| LSTM-7 → CRNN-3 → CRNN-8 | 3 | 413,393,536 | 0.01327 | 6 | 1 | 7 |
| LSTM-7 → TinyCNN-5 → CRNN-8 | 3 | 393,079,808 | 0.01608 | 5 | 3 | 8 |

Table 19: Accuracy tolerance = 0.08, Final cascades on UrbanSound8K: StabIndex, MACs, and cascade depth $K$.

| Cascade | $K$ | MACs | StabIndex | Rank(MACs) | Rank(Stab.) | Rank Sum |
|---|---|---|---|---|---|---|
| CRNN-3 → TinyCNN-5 | 2 | 118,315,264 | 0.01401 | 1 | 2 | 3 |
| LSTM-7 → CRNN-3 → TinyCNN-5 | 3 | 223,390,144 | 0.01228 | 4 | 1 | 5 |
| CRNN-3 → CRNN-8 | 2 | 120,460,800 | 0.01947 | 2 | 5 | 7 |
| TinyCNN-5 → CRNN-8 | 2 | 188,176,768 | 0.02582 | 3 | 6 | 9 |
| LSTM-7 → CRNN-3 → CRNN-8 | 3 | 225,346,304 | 0.01870 | 5 | 4 | 9 |

## E.2 FIGURES: STABILITY–COMPUTE TRADE-OFFS

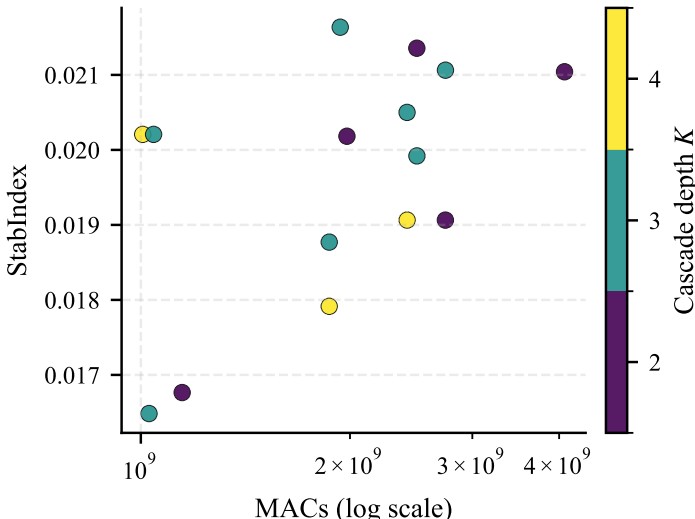

Figure 8: **Stability vs. compute on GLUE/SST-2 ($\gamma$=0.03).** Each point is a cascade; the x-axis shows expected MACs (log scale) and the y-axis shows StabIndex (lower is better). Color encodes cascade depth $K$.

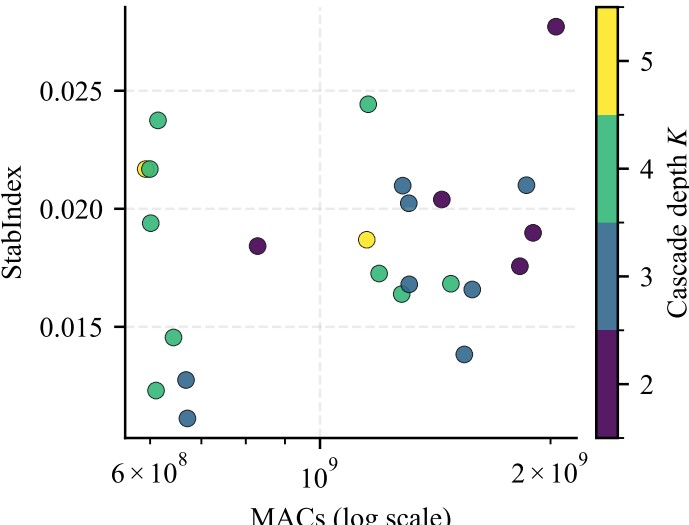

Figure 9: **Stability vs. compute on GLUE/SST-2 ($\gamma$=0.05).** Each point is a cascade; the x-axis shows expected MACs (log scale) and the y-axis shows StabIndex (lower is better). Color encodes cascade depth $K$.

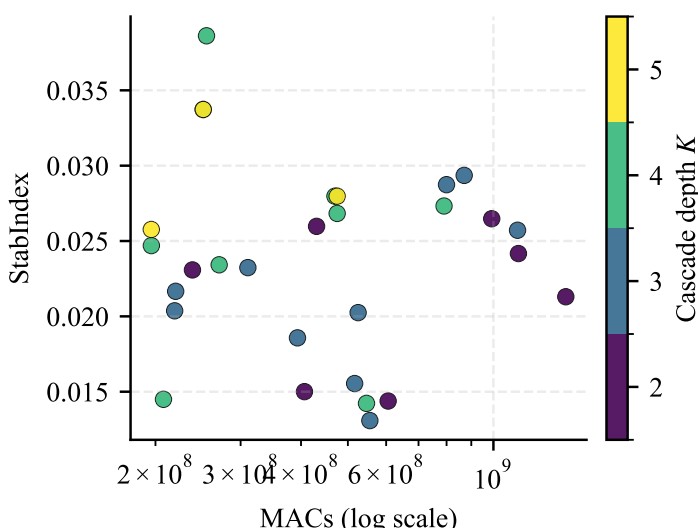

Figure 10: **Stability vs. compute on GLUE/SST-2** ($\gamma$=0.08**).** Each point is a cascade; the x-axis shows expected MACs (log scale) and the y-axis shows $\mathrm{StabIndex}$ (lower is better). Color encodes cascade depth $K$.

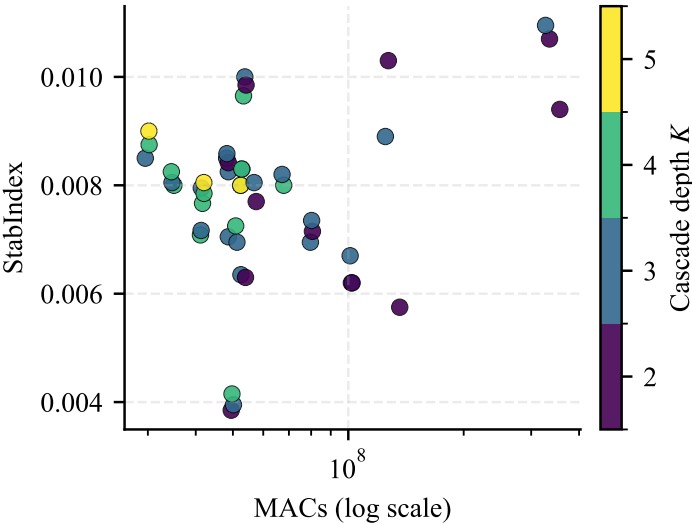

Figure 11: **Stability vs. compute on CIFAR-10** ($\gamma$=0.03**).** Each point is a cascade; the x-axis shows expected MACs (log scale) and the y-axis shows $\mathrm{StabIndex}$ (lower is better). Color encodes cascade depth $K$.

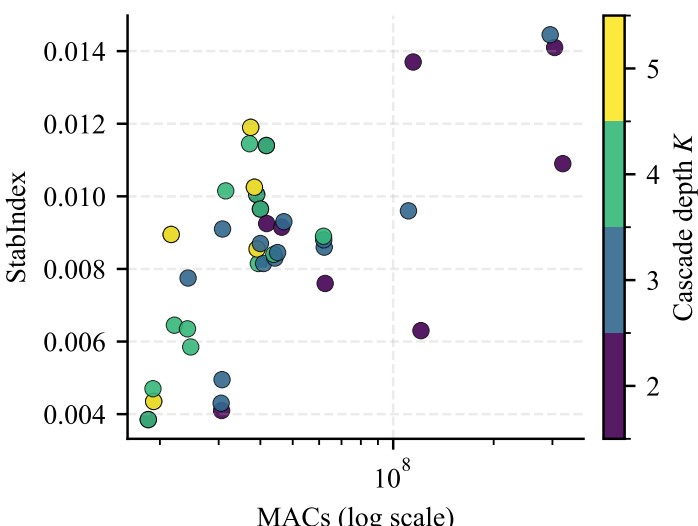

Figure 12: **Stability vs. compute on CIFAR-10** ($\gamma$=0.05). Each point is a cascade; the x-axis shows expected MACs (log scale) and the y-axis shows StabIndex (lower is better). Color encodes cascade depth $K$.

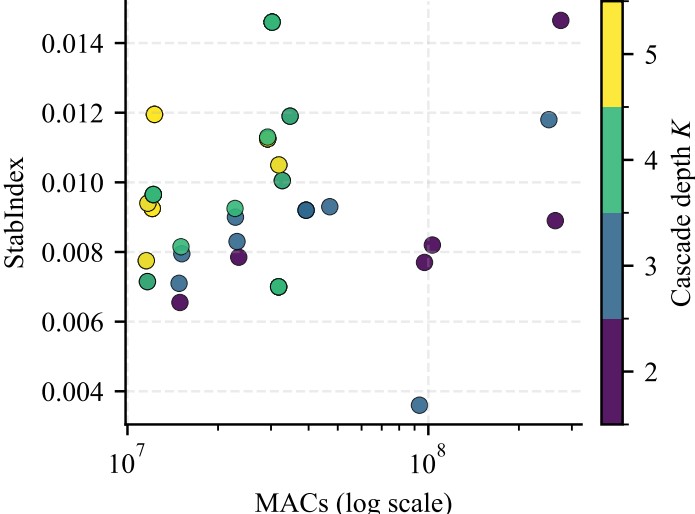

Figure 13: **Stability vs. compute on CIFAR-10** ($\gamma$=0.08). Each point is a cascade; the x-axis shows expected MACs (log scale) and the y-axis shows StabIndex (lower is better). Color encodes cascade depth $K$.

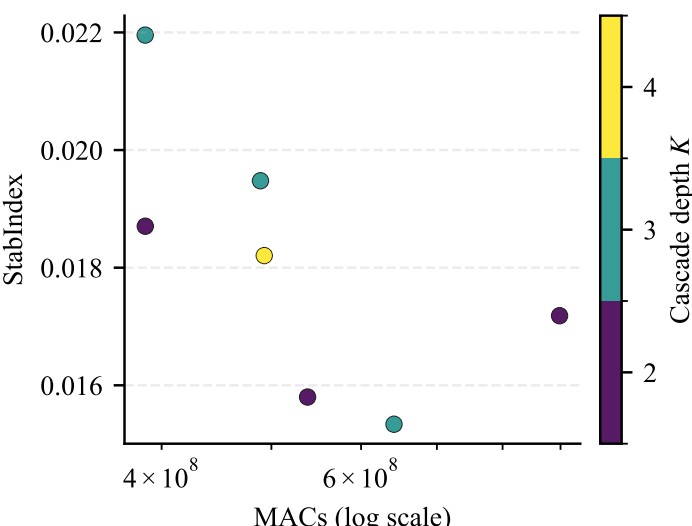

Figure 14: **Stability vs. compute on UrbanSound8K ($\gamma$=0.03).** Each point is a cascade; the x-axis shows expected MACs (log scale) and the y-axis shows $\mathrm{StabIndex}$ (lower is better). Color encodes cascade depth $K$.

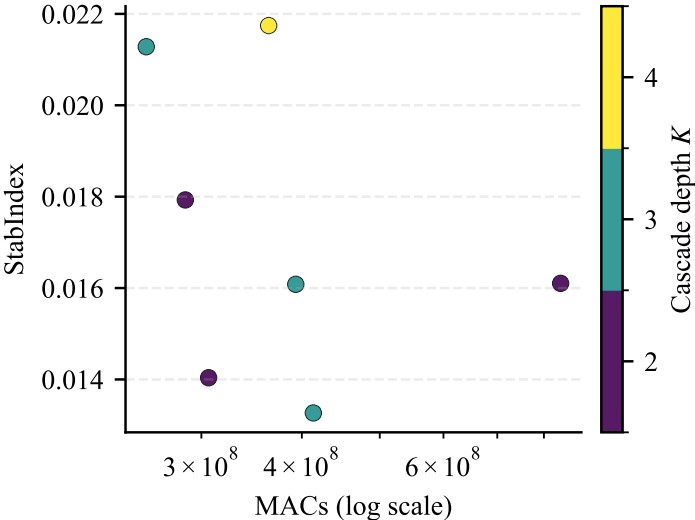

Figure 15: **Stability vs. compute on UrbanSound8K ($\gamma$=0.05).** Each point is a cascade; the x-axis shows expected MACs (log scale) and the y-axis shows $\mathrm{StabIndex}$ (lower is better). Color encodes cascade depth $K$.

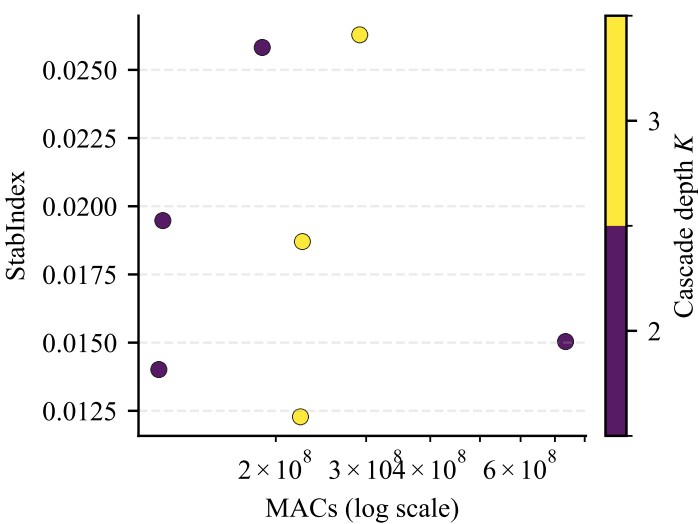

Figure 16: **Stability vs. compute on UrbanSound8K ($\gamma$=0.08).** Each point is a cascade; the x-axis shows expected MACs (log scale) and the y-axis shows $\mathrm{StabIndex}$ (lower is better). Color encodes cascade depth $K$.

