# OpenReview forum: "Stability-Aware Post-Training Cascade of Experts for Compute-Efficient Inference"
_ICLR.cc/2026/Conference — ICLR 2026 Conference Withdrawn Submission_

### Official Review · Reviewer_A8dB · 2025-10-24

**Soundness:** 1
**Presentation:** 1
**Contribution:** 2
**Rating:** 0
**Confidence:** 4

**Summary:**

The paper studies a way to form a cascade from a pool of already trained classifiers. A cascade consists of a set of classifiers arranged in a chain. A given input x goes through each classifier in order. If the current classifier is confident (i.e., its predicted class probability exceeds a threshold), then it predicts the class for x. Else, x is deferred to the next classifier in the chain. The  paper proposes an algorithm to 1. select a subset of $K$ Pareto optimal classifiers from the original pool; and 2. learn $K$ thresholds by a recurvsie threshold search scheme (Sec 2.3).

**Strengths:**

In the literature on learning to defer and model cascading, typically the deferral threshold is left as a hyperparameter. These $K$ deferral thresholds essentially control the quality and cost trade-off, and are left to the user of the system. This work is one of the few that explicitly tries to pick the right threshold. This direction has not received enough attention in the literature

**Weaknesses:**

Unfortunately, weaknesses outweigh strengths.

**Clarity**: The paper is hard to read. Insufficient clarity is my biggest concern. The paper largely focuses on the mechanical part of the algorithm or the “how”, and severely lacks explanation for “what” and “why”. To be very concrete, for instance, in Sec 2.2 (core proposal),

> We first form the strict Pareto frontier …

This is “how” you proceed with the algorithm. But “why” do you consider the Pareto frontier? The reason or motivation is missing. Same for

> L199:  We impose an upper bound on the deployment accuracy…

This is how. But “why” do you impose an upper bound? There are several other instances like this throughout Sec 2. Regarding the lack of explanation of “what”, here are some concrete examples:

> L160: On the validation set V , let $S_1(\tau) = V$ and define $A_k(\tau_k) = ..$

These are just symbols without explanation. “What” is $S_1$? and “What” is $A_k$? It took some time to digest that $A_k$ means the set of input instances that are accepted by the classifier $k$. But then I don’t see why the reader has to know this since the section after this does not use these symbols. There are instances like this throughout Sec 2.

All in all, the main section (Sec 2) lacks sufficient clarity to the point that it hinders proper evaluation of the paper’s novelty and significance. I am not sure if it is appropriate to do iterative revisions in the ICLR discussion phase.  Sec 2 needs to go through a major revision.


More examples:
* L46: “Despite their promise, existing cascaded expert systems face several obstacles”. No references at all in this section.The introduction tries to motivate that existing methods suffer from some drawbacks. But it is unclear what existing methods are being discussed. Need precise citations.

* Unclear writing in Sec 2. In Eq 1, $C$ is not defined. If $C$ is the number of classes, then it is worth making the problem setup clear up front that all the models in the pool are classifiers.

* Core section Sec 2.1 is not well written. L132: “round each softmax score to 10−3 precision and then map it onto a 10−2 -granularity level grid by packing adjacent high-score bins on the training set.”. What is “packing adjacent high-score bins on the training set”?

* In Eq 1, the significance of $s_k$ is not established.

* L140: What is $\mathcal{L}$?

* L159: If $E_{k+1}$ is the (k+1)-th expert model, then in Sec 2.2, why is $m$ used to denote a model?

* L161: “On the validation set V”. The set V should be properly defined. What does each element in V look like? Do $A_k$ and $S_k$ have to be introduced at this point? These are not immediately needed.

* Sec 2.2, L178: “We describe each model m by accuracy ACC(m)”. Is this empirical accuracy? If so, on what dataset and what data split (training, validation, or test)?


**Significance**
Experiments clearly do not have enough baselines. There is insufficient empirical evidence to support the proposed algorithm.

**Questions:**

Main suggestion: I encourage the authors to add a reason or motivation to each step of the algorithm.

Questions and concerns:
1. Experiments in Sec 3 do not have any baselines at all. Here is a natural baseline. First do the same model selection by considering the Pareto optimal models as done in the proposed approach. Then, do not learn thresholds. Set the threshold for the k-th model to be the median of the predicted scores $\tilde{s}_k(x)$. This is an ablative baseline (not a competitive one) that shows the improvement of learning the thresholds.  Have you considered this baseline? This is only one of many other baselines that are appropriate to include. The work needs several more baselines. One can change the median to a different appropriate quantile. Median is just simple to start.

2. Wisdom of Committees (https://openreview.net/forum?id=MvO2t0vbs4-) is a relevant *competitive* baseline. Consider the Pareto optimal models. At stage $k$, consider the ensemble of outputs of all models up to stage $k$ (as opposed to only considering the output of the k-th model).

3. Just a comment. Selecting only Pareto optimal models does not always lead to the best cascade. Consider a simple setup of only 2 models: $m_1$ and $m_2$. Suppose the dataset has 100 examples. $m_1$ is correct on all first 50 examples (and wrong on all others). $m_2$ is correct on examples 51 to 99, and is wrong on the 100-th example. So, ACC($m_1$) = 50% and ACC($m_2$) = 49%. Suppose $m_1$ and $m_2$ have the same MAC (cost). Your proposed model filter procedure would remove $m_2$ from the cascade. However, clearly, smartly deferring examples to these two models can get you 99% accuracy.
This is just a comment. The authors need not change the procedure. I only want to point out a drawback.  This drawback is perfectly acceptable if it can be made clear. Unfortunately, I think the paper lacks discussion on this point.

4. It would also be good to quantitatively compare the “relative” performance of the constructed cascade to an oracle cascade (i.e., the best possible cascade). In the current revision, the average accuracy and average cost of your cascade are reported without an anchor point. It’s hard to interpret how good these numbers are. There is no single right way to do this. One way is to consider all possible $K$ threshold combinations. This will trace out a region in the accuracy-cost plane. See Fig 2c in the RouterBench paper (https://openreview.net/pdf?id=IVXmV8Uxwh) for what I mean. We can then consider only the upper envelope of this region, which defines the Pareto optimal curve. Your cascade will yield one point on this plane. You could define an appropriate distance to this Pareto optimal curve to quantify the relative performance of your cascade to this oracle (here, oracle is actually a family of cascades). There is no unique way to define this distance. One could perhaps consider the Euclidean distance to the nearest point on this Pareto optimal curve. There are likely better ideas.

---

### Official Review · Reviewer_nEXb · 2025-10-30

**Soundness:** 1
**Presentation:** 2
**Contribution:** 2
**Rating:** 2
**Confidence:** 4

**Summary:**

This paper studies the problem of automatic cascade generation and proposes a novel threshold selection mechanism.

**Strengths:**

- multi-modal evaluation

- interesting idea for threshold optimization

**Weaknesses:**

- Overall a comparision against THE simple baseline is missing. Given a set of models 1) select the (eps=0) pareto models. 2) build pairwise cascades with models of adjacent complexity. (e.g. 'Efficient Inference With Model Cascades' by Lebovitz et al. in TMLR 2023 --> this paper also shows that deeper cascades have strongly diminishing returns) Please show a pareto plot of the proposed method compared against this (quality on y, cost on x). Does the method actually improve on this key plot? If yes, by how much? (e.g. given a fixed accuracy target, how much FLOPs can the method save?)

- This paper claims three key novel contributions. For two of these, evaluations are done against much too weak baselines. I will detail below, what stronger baselines can be used to have the potential of a convincing argument.

- Novelty 1: "Which base models to select". The paper proposes eps-pareto models. This causes a problem, in that now very many models become eligible to be part of a cascades (a problem that the paper then proceeds to address to some extent). However, we are not given any result that indicates to what extent it is actually helpful to use the large set of eps-pareto models, rather than the much smaller set of eps=0 i.e. just the pareto models. A sensible baseline would be 1) eps=0 and 2) eps=0 with 'evenly spaced' (in terms of size) base models (so that exhaustive search of thresholds becomes tractable) 3) eps=0 and just the largest and smallest model.

- Novelty 2: "How to optimize thresholds". The paper proposes a potentially interesting method to avoid an exhaustive search of thresholds. However, the only baseline comparison is against brute-force grid search. Simple heuristics are neglected. A sensible baseline would be setting all thresholds to the same value and sweeping that one value (yielding a method that is constant in the number of base models, rather than exponential).

- Evaluation on small problems only

- The results of model calls can be cached, so that we only need to run each model once, also when we evaluate it with multiple thresholds -- most 'calls' counted e.g. in table 2 are just cache look-ups not model evaluations.

**Questions:**

Given that model outputs can be cached (only need to be computed once and can be saved for evaluation with different thresholds), is the complexity reduction of the threshold search really that critical? Can you give some time estimates for a cache lookup?

---

### Official Review · Reviewer_CRY1 · 2025-11-02

**Soundness:** 2
**Presentation:** 3
**Contribution:** 1
**Rating:** 2
**Confidence:** 5

**Summary:**

This paper introduces a post-training cascade framework for compute-efficient inference over heterogeneous model pools. It addresses three design questions: (1) model selection via epsilon-Pareto screening, (2) threshold optimization through recursive prepend-and-reuse search, and (3) cascade ordering using cross-validated stability metrics. The cascade routes inputs through increasingly expensive models, accepting at early stages when confidence exceeds learned thresholds. The method achieves substantial compute reduction (up to 97% MACs savings) while maintaining accuracy within user-specified tolerance bounds on datasets where validation-test gaps are small (SST-2, CIFAR-10). However, on UrbanSound8K with larger validation-test distribution shift, accuracy violations exceed tolerance despite stable cross-validation.

**Strengths:**

- Practical compute efficiency: Recursive search reduces threshold optimization from combinatorial cost to polynomial cost, enabling 3-6 orders of magnitude speedup. Results show 76-97% MACs reduction when validation-test gaps are minimal.​
- Post-training black-box composition: Works with any pretrained models without retraining or architectural modification, enabling immediate deployment over existing model zoos.​
- Cross-modal evaluation: Tested on text (SST-2), vision (CIFAR-10), and audio (UrbanSound8K) with diverse architectures (116, 51, 49 models respectively), demonstrating generality.​
- Principled statistical framework: Wilson confidence intervals and tolerance adjustment provide rigor beyond naive accuracy estimates.​

**Weaknesses:**

- [most important] Limited novelty and missing baselines: Cascades, early-exit, Pareto optimization, and stability-based selection are well-known. No comparison to "Efficient Inference With Model Cascades" (TMLR 2023), which also provides all these features and much more thorough evaluations, or "Revisiting Cascaded Ensembles for Efficient Inference" (2024), both using confidence-based routing. Related work cites early papers (2018) but omits recent work. Rank-sum cascade selection is ad-hoc with no ablation versus alternatives.​
- Distribution shift failure untreated: UrbanSound8K shows 15.1pp accuracy drop versus 8pp tolerance, indicating complete method failure under validation-test mismatch. The paper acknowledges this but offers no solution, mitigation, or analysis of failure modes. No predictive indicators or robustness guarantees.​
- Shallow stability investigation: Stability index (Equation 4) averages train-validation gaps without justification. No comparison to alternative stability measures (e.g., confidence intervals from Refs, cross-validation bounds, prediction consistency). Appendix E.2 plots show stability-compute tradeoffs but lack explanation of why certain cascades are stable.​
- Insufficient ablations and analysis: No hyperparameter sensitivity analysis for recursive search ($\Delta$, W, h, T beyond K=3 calibration). No investigation of model ordering effects (why MACs-sorted versus accuracy-sorted). No study of confidence calibration impact (raw softmax is poorly calibrated). Threshold-free or learning-based routing methods not explored.
- CIFAR-10 has not been an acceptable dataset for paper-quality evaluations. Why not evaluate at least on ImageNet as the aforementioned TMLR paper?
- MACs are an OK proxy metric for cost, but it would be much better to also provide real performance measurements.

**Questions:**

- Validation-test gap mitigation strategy: Table 4 shows UrbanSound8K accuracy drops 15.1pp at epsilon=0.08, severely violating the 8pp tolerance. The reported 4.4% single-model gap versus 0.4% on CIFAR-10 indicates domain shift. Can you develop: (a) distribution shift detection heuristics (e.g., confidence histogram divergence, prediction disagreement across stages) to flag high-risk cascades before deployment, (b) adaptive reweighting or domain adaptation techniques during threshold optimization, or (c) characterization of when your method will fail? What properties of UrbanSound8K cause this failure?​
- Comparison to recent cascade baselines: How does your method compare quantitatively to "Efficient Inference With Model Cascades" (TMLR 2023 paper on Pareto-optimal cascade design) on shared benchmarks? What about "Revisiting Cascaded Ensembles for Efficient Inference" (2024 ensemble-based approach) or "Cascade-Aware Training" (2024)? These papers also optimize cascade thresholds but with different strategies; direct comparison would clarify your contribution. Further, they provide a more principled approach to selecting the models in the cascade based on their individual compute cost & quality properties. Can you provide results on ImageNet or standard benchmarks used by these works and compare them?
- Stability metric and multi-objective selection design: Your StabIndex averages train-validation gap magnitude (Equation 4). How does it compare empirically to: confidence bounds from cross-validation literature, stability selection, or prediction consistency measures? Why use rank-sum aggregation versus Pareto dominance or weighted scalarization from multi-objective optimization? Did you try other stability definitions? Can you ablate these choices on one dataset?​
- Confidence calibration and threshold robustness: You rely on uncalibrated softmax max-probability (Equation 1). Recent work shows temperature scaling or Platt scaling significantly improve early-exit robustness under distribution shift. Can you: (a) retrain threshold search with calibrated confidence estimates (temperature-scaled or post-hoc calibration), (b) compare performance on UrbanSound8K and other datasets with and without calibration, (c) investigate whether calibration reduces the validation-test gap?​
- Cascade depth and model ordering sensitivity: Your results show K=2-4 stages perform best (Tables 11-19). Is there principled guidance on optimal depth given model pool properties (accuracy gaps, compute ratios, confidence distributions)? Can you ablate model ordering (MACs-sorted versus accuracy-sorted versus confidence-sorted) and show the performance range for fixed model subsets? This would clarify whether results depend on ordering or model selection.

---

### Official Review · Reviewer_vfrS · 2025-11-02

**Soundness:** 2
**Presentation:** 3
**Contribution:** 2
**Rating:** 2
**Confidence:** 3

**Summary:**

The authors propose a mixture of experts pipeline, such that the experts are arranged in ascending order of computational demand in a cascade. The authors identify thresholds of prediction for each expert - if the satisfactory levels are met, the pipeline performs an exit, otherwise it proceeds to the next expert. For resource constrained situations, the pipeline prunes similar performance layers, based on the Wilson confidence interval. As the deeper cascade continues, the pipeline attaches and trains a lighter expert to the original well trained cascade and prunes infeasible configuration based layers. As a result, the pipeline achieves high performance with a fraction of the compute necessary.

**Strengths:**

A> The paper discusses a relevant problem about accuracy and latency trade off, and how modifying existing trained models, can help address this issue

B> The pipeline gives the user the freedom to choose between accuracy and latency. The paper demonstrates results on a wide variety of tasks.

**Weaknesses:**

A> The paper has limited novelty. Pruning on a cascade of experts has been a prevalent approach in the machine learning community, eg. [1].

B> There is limited comparison with other state of the art models. The paper demonstrates performance on three tasks, but to ensure scalability, it would be important to see if the results extend to other datasets like ImageNet for the image classification task.

C> By adding multiple models, the peak performance is dependent on the best model in the cascade. Introducing multiple models leads to choosing multiple hyperparameters cautiously, so that optimal performance is reached. Further, storing and choosing between multiple models increases the overhead for the approach. For instance, the proposed method uses 0.44 G MACs for the image classification task(91.9% on CIFAR-10). Prior work like EfficientNet, use 0.195 G MACs (single model), and demonstrate superior performance (98.1% on CIFAR 10) [2].

[1] Lu, Xudong, et al. "Not All Experts are Equal: Efficient Expert Pruning and Skipping for Mixture-of-Experts Large Language Models." Proceedings of the 62nd Annual Meeting of the Association for Computational Linguistics (Volume 1: Long Papers). 2024.
[2] Tan, Mingxing, and Quoc Le. "Efficientnet: Rethinking model scaling for convolutional neural networks." International conference on machine learning. PMLR, 2019.

**Questions:**

The paper is an interesting read.

It would be interesting to observe if the inference speed is faster than other models.

The approach is modular, so the choice of experts play an important role in the performance of the model. From a scalability perspective - would it be possible to make the pipeline work on other models and datasets too?

---

### Note · Authors · 2025-11-26

I have read and agree with the venue's withdrawal policy on behalf of myself and my co-authors.